# Training with Dynamic Sparse Heads as the Key to Effective Ensembling

## Abstract

Model ensembles have long been a cornerstone for improving generalization and robustness in deep learning. However, their effectiveness often comes at the cost of substantial computational overhead. To address this issue, state-of-the-art methods aim to replicate ensemble-class performance without requiring multiple independently trained networks. Unfortunately, these algorithms often still demand considerable compute at inference. In response to these limitations, we introduce **NeuroTrails**, a sparse multi-head architecture with dynamically evolving topology. This unexplored model-agnostic training paradigm improves ensemble performance while reducing the required parameter count. We analyze the underlying reason for its effectiveness and observe that the various neural trails induced by dynamic sparsity attain a *Goldilocks zone* of prediction diversity. NeuroTrails displays efficacy with convolutional and transformer-based architectures on vision, language, and reinforcement learning tasks. Experiments on ResNet-50/ImageNet, LLaMA-350M/C4, DQN/Atari demonstrate increased performance and stronger robustness in zero-shot generalization, while requiring significantly fewer resources.

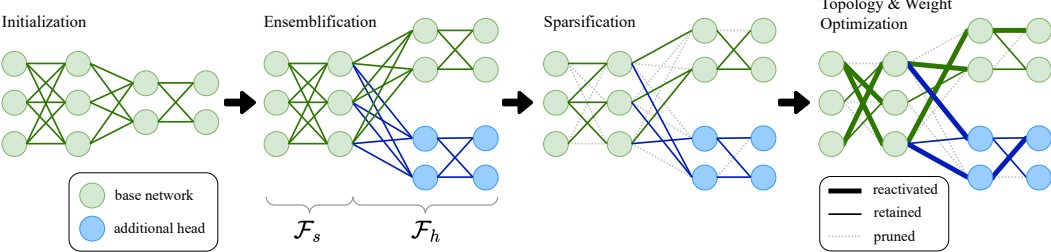

Figure 1: Illustration of NeuroTrails. We divide a network into a shared backbone $\mathcal{F}_s$ and multiple independent heads $\mathcal{F}_h$. Weights are initially pruned at random to a target sparsity ratio. Finally, the network topology is repeatedly refined through dynamic sparse training. The resulting sparse multi-head architecture achieves better performance than a full ensemble while using fewer resources.

## 1 Introduction

The idea of combining the outputs of multiple models to produce a stronger predictor has been around for a long time, with foundational works on stacking linear models (Beyer, 1981; Wolpert, 1992), bagging (Breiman, 1996) and boosting (Freund & Schapire, 1997) establishing the efficacy of this approach. Following these early developments, ensembling has proven to be a powerful technique in deep learning to increase accuracy, robustness, and generalization performance (Hansen & Salamon, 1990; Maclin & Opitz, 2011; Zhou, 2012).

A common approach involves training multiple deep neural networks independently and averaging their predictions at inference (Zhou, 2012). Random initialization allows ensemble models to explore various local optima, diversifying their predictions (Fort et al., 2020). However, the huge increase in required compute is a significant

Table 1: NeuroTrails outperforms ensembles across (self-)supervised learning and RL domains.

| Method | ResNet-50/ImageNet Accuracy (↑) | LLaMA-350M/C4 Perplexity (↓) | DQN/Atari Wins (↑) |
|---|---|---|---|
| Single Network | 76.1 | 22.8 | 1/6 |
| Full Ensemble | 77.5 | 21.3 | 0/6 |
| **NeuroTrails** | **78.1** | **20.7** | **5/6** |

disadvantage. Multiple works have attempted to reduce this overhead by, for example, factorizing weight matrices (Wen et al., 2020), network distillation (Hinton et al., 2015), or with a Multi-Input Multi-Output configuration (MIMO) (Havasi et al., 2021), usually reducing the number of parameters of an ensemble to be approximately similar to a single model. An alternative approach to reducing the parameter counts of neural networks lies in the extensive field of pruning (Frankle & Carbin, 2019; Lee et al., 2018; Wang et al., 2020) and dynamic sparse training (Mocanu et al., 2017; Evci et al., 2020). Various studies leverage these methods to address the complexity challenges associated with ensembles (Liu et al., 2022; Whitaker & Whitley, 2022).

In this paper, we approach ensembles from the perspective of TreeNet architectures (Lee et al., 2015). These are structures that share the early layers of neural networks, while retaining as many heads as a corresponding ensemble. While TreeNet's shared backbone reduces the parameter count, the performance may not always match a full ensemble, as the heads often fail to achieve enough separation in prediction diversity.

To resolve this, we introduce **NeuroTrails**, a novel training paradigm enabling ensemble models to share early backbone layers while forming diverse independent trails further in the network, see Figure 1. We train the multi-head model using dynamic sparse training, which allows NeuroTrails to adapt its network topology over time. By tuning the backbone length, the resulting model attains a *Goldilocks zone* of prediction diversity—neither too little nor too much (Section 5.2). Furthermore, the sparsity enables parameter reduction, directly translating to inference speedups (Section 5.4).

NeuroTrails is model-agnostic, outperforming ensembles built from both convolutional networks (ResNet-50, Wide-ResNet28-10, DQN) and transformer models (LLaMA-130M, LLaMA-350M). It surpasses them on vision, language, and reinforcement learning benchmarks such as Atari, CIFAR-100, ImageNet, and the Colossal Clean Crawled Corpus, see Table 1. Additionally, NeuroTrails displays strong zero-shot generalization to out-of-distribution images and downstream language tasks.

In summary, our contributions are:

- We introduce NeuroTrails, a novel training paradigm improving neural network ensembles through two key mechanisms: shared early layers and dynamic sparse training.

- We validate our model-agnostic approach with extensive vision, language, and reinforcement learning experiments on common benchmarks, showing consistent improvements.

- We provide deeper analysis on prediction diversity, real-time speedups, and key design factors—including the optimal splitting point, ensemble size, and sparsity ratio.

## 2 PRELIMINARIES

### 2.1 ENSEMBLING

Combining the strength of multiple models in an ensemble is widely studied in the literature, and has been shown to reduce variance and improve generalization (Hansen & Salamon, 1990). Ensembles can be used for uncertainty estimation (Lakshminarayanan et al., 2017), leading to more calibrated probability estimates, covering a larger portion of the problem space, bridging representation gaps left by individual models (Dietterich, 2000; Zhou, 2012). However, the additional computational cost in training and inference of neural network ensembles severely limits their scope of application (Gomes et al., 2017; Dietterich, 2000).

### 2.2 SPARSITY

The sparsification of neural networks has been a prevalent resolution to ease this computational burden (LeCun et al., 1989; Frankle & Carbin, 2019; Evci et al., 2020). Sparsifying a network involves removing a certain fraction of its parameters to create a lightweight model. Let an $n \times k$ dense layer be the weighted digraph $G = (V, E_{\text{dense}}, \boldsymbol{\theta})$ where $V = V_{\text{in}} \cup V_{\text{out}}$ is the set of neurons, $E_{\text{dense}} = V_{\text{in}} \times V_{\text{out}}$ the set of potential edges, and $\boldsymbol{\theta} \in \mathbb{R}^{nk}$ the corresponding weight matrix. A binary mask $\mathbf{m} \in \{0, 1\}^{nk}$ selects the active edge set $E = \{\, e_i \mid m_i = 1 \,\}$, producing the sparse weight matrix $\boldsymbol{\theta} \odot \mathbf{m}$. The *sparsity ratio* $S = 1 - \|\mathbf{m}\|_0 / nk \in [0, 1]$ is the fraction of edges removed.

**Pruning.** Pruning methods generally involve training a *dense* network to convergence, then selecting a mask $\mathbf{m}$ with the desired sparsity, classifying these algorithms as **dense-to-sparse**. The process ranks each weight $\theta_i$ with an importance score $s_i$, keeping the top $(1 - S)nk$ entries. Typically used scores are magnitude $s_i^{\text{mag}} = |\theta_i|$, first-order $s_i^{(1)} = |\theta_i g_i|$ with $g_i = \partial\mathcal{L}/\partial\theta_i$ (Mozer & Smolensky, 1988), and second-order $s_i^{(2)} = \frac{1}{2}\theta_i^2 H_{ii}$ with $H_{ii} = \partial^2\mathcal{L}/\partial\theta_i^2$ (LeCun et al., 1989). A short finetuning pass can restore accuracy after pruning (Han et al., 2015). See Appendix A for further background and lottery-ticket variants.

**Sparse Training.** Training neural networks with a sparse structure throughout the entire training process is the counterpart of pruning, depicting a **sparse-to-sparse** paradigm. In *static* sparse training, the network topology is fixed, making it very sensitive to the initial choice of $\mathbf{m}$. *Dynamic* sparse training (DST) solves this issue, enabling the sparse topology to be adaptive. Popular algorithms that exemplify this methodology are Sparse Evolutionary Training (SET) (Mocanu et al., 2017) and Rigged Lottery Tickets (RigL) (Evci et al., 2020). SET starts with a sparsely connected neural network and iteratively updates its structure $\mathbf{m}$ over fixed intervals $\Delta T$. At each topology update, a drop fraction $p$ of the active weights with the smallest magnitude $|\theta_i|$ is pruned, after which an equal number of inactive weights are regrown uniformly at random. RigL uses gradients of inactive connections to guide regrowth, always selecting the highest absolute gradients $|g_i|$ as most promising.

## 3 NEUROTRAILS

We introduce NeuroTrails, a novel training paradigm to enhance the performance of neural network ensembles, while reducing their parameter complexity (see Figure 1). The method is model-agnostic and can be applied to any architecture. See Appendix D for a concise pseudocode overview.

**Architecture split.** Let the base network $\mathcal{F}$ be a composition of $L$ blocks

$$\mathcal{F}(x; \boldsymbol{\theta}) = f_L\big(f_{L-1}(\cdots f_1(x; \boldsymbol{\theta}_1)\cdots; \boldsymbol{\theta}_{L-1}); \boldsymbol{\theta}_L\big)$$

where a block is a collection of neural network layers, such as a residual or transformer block. We choose a split index $1 \leq \ell \leq L$ and partition into

$$\mathcal{F}_s(x; \boldsymbol{\theta}_s) = f_\ell \circ \cdots \circ f_1, \qquad \mathcal{F}_h(x; \boldsymbol{\theta}_h) = f_L \circ \cdots \circ f_{\ell+1}.$$

We instantiate $M$ independent heads $\mathcal{F}_h^{(i)}$ $(i = 1, \ldots, M)$, each with separately initialized weights $\boldsymbol{\theta}_h^{(i)}$ and sparse mask $\mathbf{m}_h^{(i)}$. These unique initial conditions seed distinct "neural trails"—deep, long-range connectivity paths that give the multi-head network its diversity. The shared trunk $\mathcal{F}_s$ likewise carries a mask $\mathbf{m}_s$. We analyze the ideal backbone length $\ell$ in Section 5.1, and investigate the effect of different sparsity ratios $S$ in Appendix H. In the remainder of this paper, we will denote the number of blocks in the backbone and heads by $|\mathcal{F}_s| = \ell$ and $|\mathcal{F}_h| = L - \ell$, respectively.

**Training.** On a minibatch $(x, y)$, we compute each head's logits

$$\hat{y}^{(i)} = \mathcal{F}_h^{(i)}(F_s(x; \boldsymbol{\theta}_s); \boldsymbol{\theta}_h^{(i)}).$$

Individual losses $\mathcal{L}_i$ for each head $i$ are calculated and averaged to form the composite loss,

$$\mathcal{L}(\Theta) = \frac{1}{M}\sum_{i=1}^{M}\mathcal{L}_i\big(\hat{y}^{(i)}, y\big), \quad \Theta = (\boldsymbol{\theta}_s, \boldsymbol{\theta}_h^{(1)}, \ldots, \boldsymbol{\theta}_h^{(M)}),$$

which is used to update all active parameters through a masked version of stochastic gradient descent (Robbins & Monro, 1951) or Adam (Kingma & Ba, 2015). Every $\Delta T$ steps, each component (shared or head $i$), performs a topology update through dynamic sparse training. This process consists of (1) layerwise pruning of $p$ weights, and (2) reinitializing an equal number $p$ previously inactive connections, thereby maintaining a constant density $\|\mathbf{m}\|_0/nk$ while exploring new sparse trails.

In computer vision experiments, we reactivate weights with RigL (Evci et al., 2020) and prune by standard magnitude $|\theta_i|$, as recommended by Nowak et al. (2023). We use the Erdős–Rényi (ER) approach (Mocanu et al., 2017; Evci et al., 2020) to distribute the global sparsity $S$ into layerwise

sparsity ratios. ER has been shown to yield superior performance over simply setting each layer's sparsity to $S$, i.e., uniform sparsity (Liu et al., 2023). In a nutshell, ER assigns higher sparsity ratios to larger layers. See Appendix A for additional information.

For language modeling, we likewise use ER, but leave attention projections dense while sparsifying all other layers. Furthermore, we also use RigL for growth, but we prune using *soft magnitude*, shown to work well for language models by Zhang et al. (2025). In this procedure, a weight's absolute value determines a *probability* of being pruned, instead of simply pruning the smallest weights.

Dense models tend to overfit once training is prolonged, whereas sparse networks keep improving as they are still refining both weights and topology (Liu et al., 2021b). According to the schedules of Evci et al. (2020), we extend the training of sparse variants by at most $1/(1-S)$, always keeping the total number of floating-point operations (FLOPs) for training below those of their dense counterparts. Exact number of epochs—or updates in the case of language modeling—appear in Appendix E.

**Inference.** During inference, the final prediction is computed through soft voting, averaging logits across all ensemble members:

$$\bar{y} = \frac{1}{M} \sum_{i=1}^{M} \mathcal{F}_h^{(i)}(F_s(x; \boldsymbol{\theta}_s); \boldsymbol{\theta}_h^{(i)}).$$

The shared backbone $F_s(x; \boldsymbol{\theta}_s)$ forward pass naturally only needs to be computed once. NeuroTrails ensures that while ensemble members share early feature extractors, the heads develop distinct predictive pathways through sparse connectivity patterns, thereby stimulating diversity.

## 4 EXPERIMENTS

We compare our methods against a single model, a full ensemble, and various state-of-the-art efficient ensemble methods in the literature, including MIMO (Havasi et al., 2021), TreeNet (Lee et al., 2015), Batch Ensemble (Wen et al., 2020), as well as DST and EDST ensembles (Liu et al., 2022). See Section 6 for detailed descriptions of these baselines. All architectures use the following base models: Wide-ResNet28-10 on CIFAR-100, ResNet-50 on ImageNet, and LLaMA-130M/350M on C4. Details on the training regime and hyperparameters are shared in Appendix E.

For computer vision experiments, we report the mean test accuracy, negative log-likelihood (NLL), and expected calibration error (ECE). In language modeling, our main metric is perplexity on the C4 validation set. We include the required number of FLOPs for training and inference. Next to the name of the model, we indicate the ensemble size (or number of heads) $M$ and sparsity ratio $S$. See Appendix F for further details on the metrics.

### 4.1 COMPUTER VISION

As shown in Tables 2 and 3, NeuroTrails demonstrates strong performance both on CIFAR-100 and ImageNet, while using significantly fewer FLOPs at inference time. We present additional results on Tiny-ImageNet in Appendix G. The low FLOPs required at inference are crucial, making NeuroTrails a compelling choice for deployment in resource-constrained environments. See Section 5.4 for the real-time speedups that are directly available.

**Robustness against Corruptions.** To test Neuro-Trails for its zero-shot generalization capability, we evaluate its robustness on ImageNet-C, a dataset of corrupted ImageNet samples with various severity levels (Hendrycks & Dietterich, 2019). Furthermore, we test on ImageNet-Sketch (Wang et al., 2019), a collection of black-and-white sketched illustrations, assessing the model's ability to extrapolate to out-of-domain (OOD) data. The results in Figure 2 show that NeuroTrails consistently outperforms the full ensemble across all severity levels and tasks, while requiring a fraction of its total FLOPs.

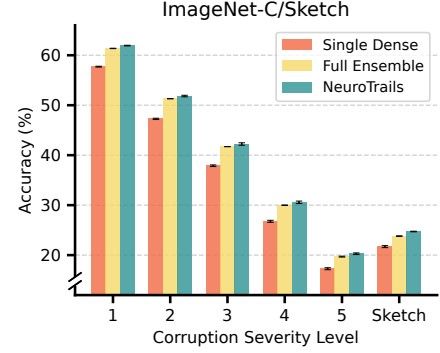

Figure 2: Zero-shot generalization ability.

Table 2: Performance on **CIFAR-100** with Wide-ResNet28-10 as the base. NeuroTrails and TreeNet have 8 blocks in the heads, with 4 remaining blocks in the shared backbone. Results marked with * are from Havasi et al. (2021), ** from Liu et al. (2022), and *** from Lee & Lee (2024).

| Method | Accuracy (↑) | NLL (↓) | ECE (↓) | Train FLOPs (↓) | Infer. FLOPs (↓) |
|---|---|---|---|---|---|
| Single Dense * | 79.8 | 0.875 | 0.086 | 3.6e17 | 10.5e9 |
| MIMO ($M = 3$) * | 82.0 | 0.690 | 0.022 | 1.00× | 1.00× |
| EDST Ensemble ($M = 7$) ($S = 0.9$) ** | 82.6 | 0.653 | 0.036 | **0.57×** | 1.17× |
| DST Ensemble ($M = 3$) ($S = 0.8$) ** | 83.3 | **0.623** | **0.018** | 1.01× | 1.01× |
| Batch Ensemble ($M = 4$) * | 81.5 | 0.740 | 0.056 | 1.10× | 1.10× |
| NFE ($M = 3$) *** | 83.5 | 0.658 | 0.061 | 1.02× | 1.02× |
| TreeNet ($M = 3$) | 83.2 | 0.673 | 0.052 | 2.91× | 2.91× |
| Full Ensemble ($M = 3$) | 83.3 | 0.663 | 0.042 | 3.00× | 3.00× |
| NeuroTrails ($M = 3$) ($S = 0.8$) | 83.8 | 0.681 | 0.044 | 0.85× | 0.47× |
| NeuroTrails ($M = 5$) ($S = 0.9$) | **83.9** | 0.675 | 0.041 | 0.67× | **0.37×** |

Table 3: Performance on **ImageNet** with ResNet-50 as the baseline model. NeuroTrails and TreeNet have 10 blocks in their multi-head structure, with 6 remaining blocks in the shared backbone. Results marked with * are from Havasi et al. (2021) and ** from Liu et al. (2022).

| Method | Accuracy (↑) | NLL (↓) | ECE (↓) | Train FLOPs (↓) | Infer. FLOPs (↓) |
|---|---|---|---|---|---|
| Single Dense * | 76.1 | 0.943 | 0.039 | 4.8e18 | 8.2e9 |
| MIMO ($M = 2$) ($\rho = 0.6$) * | 77.5 | 0.887 | 0.037 | 1.00× | 1.00× |
| EDST Ensemble ($M = 4$) ($S = 0.8$) ** | 77.7 | 0.935 | 0.064 | **0.87×** | 1.78× |
| DST Ensemble ($M = 2$) ($S = 0.8$) ** | **78.3** | 0.914 | 0.060 | 1.12× | 1.12× |
| Batch Ensemble ($M = 4$) * | 76.7 | 0.944 | 0.049 | 1.10× | 1.10× |
| TreeNet ($M = 3$) | 78.1 | 0.886 | 0.053 | 2.91× | 2.91× |
| Full Ensemble ($M = 4$) * | 77.5 | 0.877 | **0.031** | 4.00× | 4.00× |
| NeuroTrails ($M = 3$) ($S = 0.7$) | 78.1 | **0.861** | 0.038 | 1.10× | **0.67×** |

## 4.2 LANGUAGE MODELING

We pretrain variants of LLaMA-130M and LLaMA-350M on the *Colossal Clean Crawled Corpus* (Raffel et al., 2020, C4). Motivated by the work of Wu et al. (2025), we use a low sparsity ratio in these experiments, but maintain the adaptive nature of dynamic sparse training. The results in Table 4 show that NeuroTrails performs strongly on transformer architectures, achieving the best validation perplexity. Despite using a lower sparsity ratio in the language domain, our algorithm yields a lightweight model with lower inference FLOPs than both TreeNet and the full ensemble.

**Evaluation on Downstream Tasks.** We test our pretrained LLAMA-350M models for zero-shot generalization to multiple downstream tasks. The results in Table 5 compare model accuracy across seven benchmarks: MMLU (Hendrycks et al., 2021), BoolQ (Clark et al., 2019), ARC (Clark et al., 2018), PIQA (Bisk et al., 2019), Hellaswag (Zellers et al., 2019), OpenbookQA (Mihaylov et al., 2018), and WinoGrande (Sakaguchi et al., 2019). These tasks span multiple domains including common sense reasoning, multiple choice question answering, and scientific knowledge. NeuroTrails achieves the highest average accuracy, suggesting that it offers improved generalization and robustness across a wide range of language tasks.

Table 4: Pretraining performance on the **C4** dataset with LLaMA-130M/350M as the baseline model. NeuroTrails and TreeNet use ⅔ of the transformer blocks in the heads, with ⅓ in the backbone.

| Method | Perplexity (↓) | Training FLOPs (↓) | Inference FLOPs (↓) |
|---|---|---|---|
| *LLaMA-130M* | | | |
| Single Dense | 29.06 | 3.5e18 | 2.2e11 |
| TreeNet ($M = 3$) | 26.46 | **2.21×** | 2.21× |
| Full Ensemble ($M = 3$) | 26.88 | 3.00× | 3.00× |
| NeuroTrails ($M = 3$) ($S = 0.1$) | **26.00** | **2.21×** | **1.99×** |
| *LLaMA-350M* | | | |
| Single Dense | 22.80 | 4.2e19 | 6.9e11 |
| TreeNet ($M = 3$) | 21.06 | **2.27×** | 2.27× |
| Full Ensemble ($M = 3$) | 21.25 | 3.00× | 3.00× |
| NeuroTrails ($M = 3$) ($S = 0.1$) | **20.70** | **2.27×** | **2.04×** |

Table 5: Zero-shot accuracy (↑) of various LLaMA-350M models across seven downstream tasks.

| Method | MMLU | BoolQ | ARC | PIQA | Hellaswag | OBQA | WinoGrande | **Avg.** |
|---|---|---|---|---|---|---|---|---|
| Single Dense | 22.92 | 58.47 | 40.24 | 62.51 | 28.31 | 13.60 | **52.49** | 39.79 |
| TreeNet ($M = 3$) | **22.97** | 58.65 | 40.40 | 62.95 | **28.45** | 15.00 | 51.30 | 39.96 |
| Full Ensemble ($M = 3$) | **22.97** | 58.23 | 40.36 | 62.68 | 28.18 | 14.40 | 51.70 | 39.78 |
| NeuroTrails ($M = 3$) ($S = 0.1$) | 22.92 | **60.49** | **41.71** | **63.28** | 28.43 | **15.80** | 50.51 | **40.45** |

## 4.3 REINFORCEMENT LEARNING

We extend the applicability of NeuroTrails to the field of reinforcement learning (RL). In this experiment, we take a standard Deep Q-Network (DQN) (Mnih et al., 2013) and adjust its architecture to either NeuroTrails, TreeNet, or a Full Ensemble. Similar to earlier experiments, we ensure that each head is trained independently. However, in RL the data (i.e., experience) needs to be gathered by the agent itself. We decide to take actions after averaging Q-values across heads, meaning heads jointly take decisions, but are independently trained on data from the replay buffer. We train for 10M steps (40M frames) on six Atari environments (Bellemare et al., 2013), and report the interquartile mean (IQM) over 8 seeds. As shown in Table 6, NeuroTrails performs well accross the environments, even with a relatively high sparsity level of 80%. See Appendix E for further experimental details.

Table 6: Reinforcement learning return (↑) on six Atari environments with DQN as the base model. We train for 10M env steps and report IQM ± s.e.m. over 8 seeds, following Agarwal et al. (2021).

| Method | Asterix | BeamRider | Breakout | Seaquest | SpaceInvaders | UpNDown |
|---|---|---|---|---|---|---|
| Single Dense | 3200.7 ±368.9 | 4201.7 ±77.1 | 126.2 ±71.7 | 605.0 ±124.4 | **632.0** ±36.1 | 6202.7 ±224.3 |
| TreeNet ($M = 3$) | 3010.1 ±461.4 | 4988.9 ±35.9 | 268.0 ±12.4 | 123.5 ±6.6 | 478.9 ±42.4 | **7241.6** ±471.3 |
| Full Ensemble ($M = 3$) | 4698.9 ±403.5 | 5072.3 ±56.5 | 223.8 ±3.2 | 263.7 ±83.7 | 539.1 ±63.7 | 6335.1 ±149.2 |
| NeuroTrails ($M = 3$) ($S = 0.8$) | **6058.5** ±142.8 | **5742.2** ±130.4 | **284.6** ±6.9 | **2331.9** ±108.2 | **626.7** ±131.6 | 6583.9 ±292.8 |

## 5 ANALYSIS

In this section, we explore various design choices for NeuroTrails. All results reported here were obtained using Wide-ResNet28-10 on CIFAR-100, and present the mean and standard deviation over 3 independent seeds. For additional analysis on the effect of different sparsity ratios, see Appendix H.

### 5.1 BACKBONE LENGTH

An essential hyperparameter of Neuro-Trials is the optimal split index $l$. The ideal architecture may depend on both the sparsity ratio $S$ and the number of heads $M$; our analysis focuses on the configuration with 80% sparsity and 3 heads. In addition, we examine different sparsification methods on the same plot.

As detailed in Section 3, we split the architecture between blocks, where each block in Wide-ResNet28-10 consists of two convolutional layers, two batch normalization layers and a residual connection. The base network Wide-ResNet28-10 has 12 blocks in total, so we can vary the backbone length across this depth. The results shown in Figure 3 reveal that performance is maximized most efficiently with a split point at 8

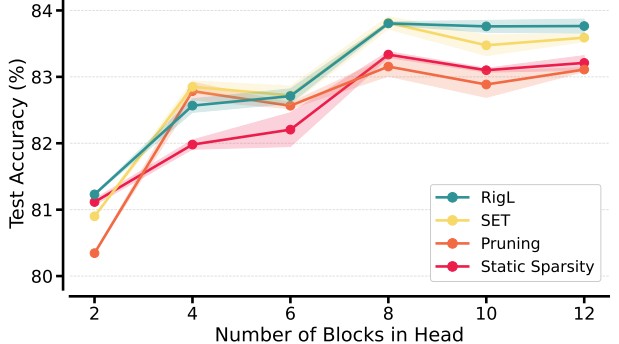

Figure 3: NeuroTrails models with varying backbone sizes and sparsification methods (on CIFAR-100 with Wide-ResNet28-10). **Backbone Length:** The most *effective* (optimizing accuracy and efficiency) backbone length appears around 1/3 of the network, meaning 8/12 blocks in head. **Sparsification:** RigL and SET demonstrate superior performance, confirming DST as the optimal approach.

blocks per head. This architecture consists of four shared backbone blocks ($|\mathcal{F}_s| = 12 - 8 = 4$) and eight blocks for each of the independent heads ($|\mathcal{F}_h| = 8$), resulting in approximately one-third of the network serving as the shared backbone. The different sparsification methods used have varying performance. However, the dynamic nature of RigL and SET helps them to consistently surpass static sparse training and standard one-time pruning.

## 5.2 PREDICTION DIVERSITY

We analyze the effect of different NeuroTrails settings on the prediction diversity and its performance. Although numerous metrics exist for quantifying diversity (Kuncheva & Whitaker, 2003), we adopt prediction disagreement (PD), one of the most widely used. PD is defined as the proportion of test samples where ensemble members produce conflicting predictions (Skalak, 1996).

Analysis of PD patterns in Table 7 reveals a monotonic increase in inter-head disagreement as the proportion of the NeuroTrails architecture allocated to independent heads grows. This observation aligns with intuition: As a larger portion of the network is dedicated to the heads, the extra head-only layers let each branch specialize, so their outputs drift further from the initially shared representation. A surprising finding emerges from our most accurate configuration with $|\mathcal{F}_h| = 8$: This model exhibits *lower* prediction disagreement between heads (14.6) compared to a full ensemble (15.4) and configurations with more blocks in the heads (up to 16.0), while being superior in performance.

This observation points to the existence of an optimal disagreement threshold, which we refer to as the *PD Goldilocks zone* (due to the amount being 'just right'). Beyond this threshold, excessive prediction diversity among ensemble members begins to degrade model performance. When heads make significantly divergent predictions for the same input, they cease to complement each other and instead compete, negating their contributions. This insight highlights that, while a certain level of diversity is beneficial in ensemble learning, excessive diversity can be detrimental, see Figure 4. Achieving the right balance between diversity and consensus is essential to maximize ensemble performance. For further analysis on this issue, see Appendix J.

**Prediction Disagreement over time.** We observe in Figure 5 that PD decreases throughout training as accuracy is growing for NeuroTrails ($M=3$, $S=0.8$). At initialization PD is relatively high ($30 \sim 40\%$), continues to decrease before reaching a steady value of approximately 14.6% at the end of the training. The relationship between PD and accuracy exhibits a notable negative correlation, particularly evident at transition points of the stepwise learning rate decay. This analysis highlights that while high diversity between heads does not guarantee better performance, low diversity similarly limits ensemble benefits.

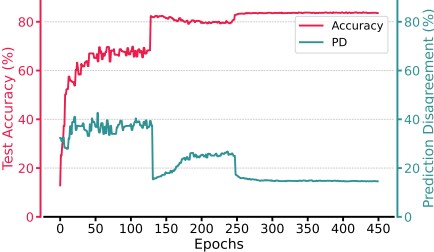

Figure 5: Accuracy and Prediction Disagreement over time for NeuroTrails on CIFAR-100, displaying an inverse trend.

Table 7: Comparing prediction disagreement (PD) and test accuracy on CIFAR-100. NeuroTrails achieves peak accuracy at $|\mathcal{F}_h| = 8$, with lower PD than configurations using more head blocks. This suggests that optimal performance lies in a *Goldilocks zone* where PD is neither too low nor too high.

| Blocks in head | PD (%) | Accuracy (%) |
|---|---|---|
| 2 | $2.9 \pm 0.17$ | $80.89 \pm 0.01$ |
| 4 | $11.2 \pm 0.41$ | $82.85 \pm 0.09$ |
| 6 | $12.4 \pm 0.28$ | $82.71 \pm 0.14$ |
| 8 | $14.6 \pm 0.36$ | $\mathbf{83.81 \pm 0.10}$ |
| 10 | $15.3 \pm 0.12$ | $83.47 \pm 0.16$ |
| 12 | $16.0 \pm 0.06$ | $83.59 \pm 0.08$ |
| Full Ensemble | $15.4 \pm 0.34$ | $83.62 \pm 0.10$ |

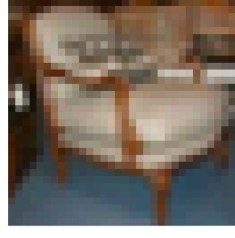

**NeuroTrails** (8 blocks)
Predictions: couch, chair, chair
Aggregate: chair

**NeuroTrails** (12 blocks)
Predictions: chair, bed, table
Aggregate: bed

Ground truth: chair

Figure 4: Illustration of a CIFAR-100 test-set image where too much prediction diversity between heads degrades performance. NeuroTrails with 8 blocks in each head seems to get the amount of diversity *just right* for optimal performance. More examples appear in Appendix J.

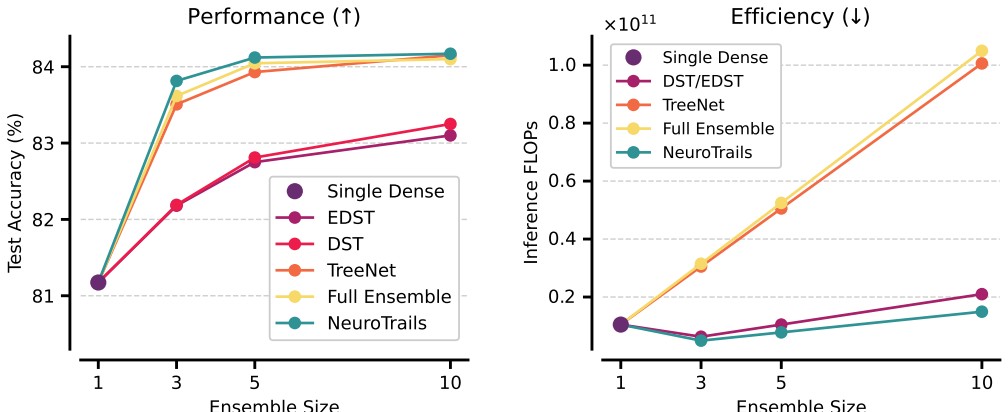

Figure 6: Effect of the ensemble size on CIFAR-100 with Wide-ResNet28-10. NeuroTrails achieves higher accuracy than a full ensemble (**left**) while consuming only a fraction of the FLOPs (**right**).

## 5.3 ENSEMBLE SIZE

We analyze the impact of ensemble size $M$ on performance and efficiency. Single networks and standard ensembles are fully dense, while NeuroTrails uses 80% sparsity and $|\mathcal{F}_h| = 8$ across all experiments in this section. The results are summarized in Figure 6. Both traditional ensembles and NeuroTrails show significant accuracy gains as the ensemble size increases from 1 to 10, with NeuroTrails consistently outperforming the baselines across all sizes. The most substantial improvements occur between sizes 1 and 3, followed by diminishing returns.

The trade-offs are further illustrated in the right plot of Figure 6. Due to its high sparsity, NeuroTrails incurs significantly lower computational costs, scaling more efficiently with ensemble size. These gains could be further amplified—e.g., a 5-head NeuroTrails can support 90% sparsity without a drop in performance on CIFAR-100 (see Table 2)—while larger ensembles may enable even greater sparsity. Exploring such configurations is a promising avenue for future work.

## 5.4 INFERENCE SPEEDUP

While FLOPs reduction is a widely used proxy for model efficiency, achieving real-world speedups often hinges on hardware compatibility and software execution paths. Recent hardware advances, such as the Cerebras CS-2 system, have shown that unstructured sparsity can translate into substantial runtime performance gains, even on GPU-class accelerators (Cerebras, 2024).

In parallel, software frameworks such as DeepSparse already deliver substantial inference-time speedups on commodity CPU hardware (NeuralMagic, 2021). In our experiments with CIFAR-100 and $M$=3, we observe that NeuroTrails models significantly outperform full ensembles in terms of practical efficiency, see Figure 7. For example, NeuroTrails ($S$=0.95) achieves a throughput of 53.16 images per second, similar to a single dense model, while achieving much higher accuracy. These results position NeuroTrails on the Pareto front of efficiency and performance, giving a compelling solution for deployment on smartphones or other edge devices, where resources are constrained and GPUs are often unavailable. Software frameworks for unstructured sparsity on GPUs are likewise on the horizon; details are described in Appendix I.

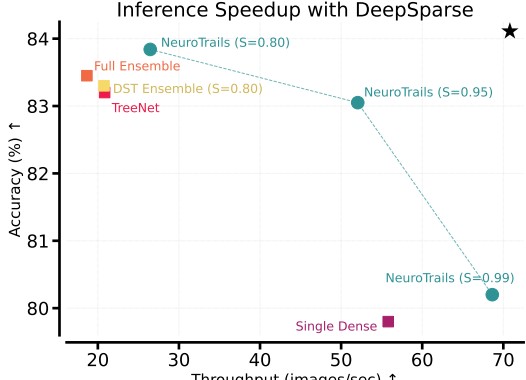

Figure 7: NeuroTrails forms the Pareto front of efficiency and performance on CPU hardware with DeepSparse. Measuring Wide-ResNet28-10 on CIFAR-100 with $M$=3 for all ensembling methods.

Table 8: Comparison between baselines across key ensembling desiderata. NeuroTrails exhibits a unique combination. Symbols: ✓ = meets criterion, × = does not meet, ∼ = partially.

| Method | Prediction Diversity | Efficient Inference | Low Training FLOPs | High Performance |
|---|---|---|---|---|
| Single Dense | × | ✓ | ✓ | × |
| MIMO | ✓ | ∼ | ✓ | ∼ |
| EDST Ensemble | ∼ | ∼ | ✓ | ∼ |
| DST Ensemble | ✓ | ∼ | ∼ | ∼ |
| TreeNet | ∼ | ∼ | ✓ | ∼ |
| Full Ensemble | ✓ | × | × | ✓ |
| **NeuroTrails** | ✓ | ✓ | ✓ | ✓ |

## 6 RELATED WORK

In deep learning, multiple attempts have been made to achieve ensemble-level performance while attaining significant reductions in parameter count and FLOPs. In Table 8 we provide a direct comparison between NeuroTrails and the varying baselines used in our experiments, indicating that our method presents a novel combination of characteristics. We focus on the most relevant methods in this section; Appendix A expands on additional topics, including Mixture-of-Experts, the Lottery Ticket Hypothesis, and further sparse training studies.

Batch Ensemble (Wen et al., 2020) introduced an efficient ensemble approach by decomposing the ensemble members into a shared matrix and rank-one personalized matrices, achieving near-single network computational costs. Multi-Input Multi-Output Ensembles (MIMO) (Havasi et al., 2021) subsequently improved on this method by ensembling only input and output layers, demonstrating enhanced performance across ensemble architectures. In MIMO, the full original network is always used as the main structure, while adding heads as additional layers at the input and output ends. NeuroTrails differs in this regard, as it only splits into heads on the output side. Furthermore, NeuroTrails has the ability to flexibly configure where our backbone splits into multiple heads, not needing to keep the full original network intact.

In TreeNets, Lee et al. (2015) propose sharing early layers for ensembles. We enhance this approach with two major components: (1) by incorporating dynamic sparse training, which fosters greater diversity and independence among neural pathways throughout the multi-headed network, significantly reducing the number of parameters and FLOPs required, particularly during inference; and (2) by splitting the backbone based on layer-based blocks rather than individual layers, preserving the structural integrity inherent in widely-used architectures such as ResNets (He et al., 2016) and Transformers (Vaswani et al., 2017).

Liu et al. (2022) use dynamic sparse training for ensembles, but do not use a multi-headed network structure. In the DST ensemble approach, independent sparse neural networks are trained from scratch, while their Efficient-DST (EDST) method creates an ensemble from a single network by using distinct model checkpoints throughout training.

## 7 CONCLUSION

We propose **NeuroTrails**, a novel training paradigm that is straightforward to integrate into various neural network architectures. The methodology splits a network into multiple sparse heads, optimizing their topology through dynamic sparse training. Extensive experiments demonstrate significant improvements across supervised, self-supervised, and reinforcement learning settings, alongside lower inference FLOPs and practical CPU throughput gains. NeuroTrails reveals that ensembling all layers is not a necessary condition to achieve optimal performance. Early-stage representation learning is more effectively handled through a single sparse backbone.

Our analysis highlights a *Goldilocks* zone of prediction disagreement: too little diversity wastes ensemble benefits, too much disrupts aggregation. The backbone-to-heads splitting point provides a simple, general knob to repeatedly place models near this sweet spot. More broadly, our results suggest a reframing of ensembling: share early, grow sparse, and control diversity, serving as core design principles for achieving scalable accuracy, robustness, and efficiency.

ETHICS STATEMENT

This work advances core machine learning capabilities by improving the performance and efficiency of neural networks. While we focus on algorithmic improvements, we acknowledge that, like most technical advances in ML, this work may have various societal impacts. We encourage thoughtful consideration of these implications when building upon this research.

By reducing parameter counts and inference FLOPs, NeuroTrails enables more efficient neural networks that are easily deployable in resource-constrained environments. These improvements help lower computational overhead and support broader, more sustainable use of AI technologies, especially as hardware increasingly catches up to exploit the use of unstructured sparsity.

REPRODUCIBILITY

We provide our source code in the Supplementary Material and will make it publicly available at camera-ready. The algorithm's implementation is described in Section 3 and Appendix D. Further training settings, hyperparameters, and model architectures are described in Appendices B and E.

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

APPENDIX

TABLE OF CONTENTS

## A    EXTENDED BACKGROUND AND RELATED WORK

### A.1    ENSEMBLING

**Learning Dynamics.** There are multiple works that investigate how training procedures shape ensemble diversity. Webb et al. (2019) show that fully end-to-end (joint) training of multi-branch ensembles can fail in over-parameterized regimes, with the optimum often lying between independent and joint training. Extending this, Jeffares et al. (2023) identify learner collusion under joint training, where members co-adapt to the shared loss and lose predictive diversity. Abe et al. (2024) demonstrate that more diversity is not automatically beneficial: certain forms of predictive disagreement can be pathological, harming accuracy and calibration. Finally, Wood et al. (2023) provide a formal bias–variance–diversity decomposition across common losses, reframing ensemble design as managing a three-way trade-off rather than maximizing diversity. These insights align with our *Goldilocks* view: avoid collusion from excessive joint training, but also avoid too much diversity when that becomes counterproductive.

**Mixture-of-Experts.** Mixture-of-Experts (MoE) models (Jacobs et al., 1991), such as Switch Transformers (Fedus et al., 2021), also attach multiple expert subnetworks to a shared backbone, but their goal is conditional computation. A learned router selects one or a few experts per token, so only a fraction of the heads run on each forward pass; afterward, the router re-weights and merges the expert outputs. In contrast, NeuroTrails does not need to train a router and simply activates every sparse head. We do not re-merge intermediate activations and aggregate only at the final logits stage. The absence of routing simplifies training, eliminates token-level gating hyperparameters, and ensures deterministic FLOPs, while dynamic sparse heads keep total compute low and enable efficient and parallelizable inference.

**Network Fission Ensembles.** Network Fission Ensembles (NFE) propose an ensemble learning approach that transforms a conventional neural network into a multi-exit structure through weight pruning and balanced weight grouping (Lee & Lee, 2024). A key advantage of NFE is that it does not require widening layers; all gains are made through intricate arranging of existing layers and parameters. NeuroTrails takes a fundamentally different approach. Our method strategically initializes independent copies of specific network layers into a multi-head structure, followed by sparsification and dynamic topology update. This approach more closely resembles traditional ensemble architectures in its behavior.

**Ensembling in RL.** Within reinforcement learning, ensembling-like approaches have proven to be a powerful tool to stabilize learning or improve sample-efficiency. In algorithms such as Double-DQN (Hasselt et al., 2015), SAC (Haarnoja et al., 2018) and TD3 (Fujimoto et al., 2018) the minimum of two critic networks is used to prevent overestimation of Q-values. Even more than two critics seems to work well (Chen et al., 2021), and ensembling the ouput of multiple RL algorithms (Wiering & van Hasselt, 2008) can be beneficial. Averaging past target networks can help stabilize learning (Anschel et al., 2017).

## A.2 SPARSITY

**Lottery Ticket Hypothesis.** There exists a family of sparsification methods based on the Lottery Ticket Hypothesis (Frankle & Carbin, 2019), which stipulates that each randomly initialized network already contains a subnetwork that can be as accurate as the full network when trained in isolation. Search of this subnetwork through Iterative Magnitude Pruning (IMP) involves training the dense network, pruning $p$ fraction of weights based on magnitude, and resetting weights to their initial states (excluding the already pruned parameters). Subsequent works have refined and significantly improved this process (Zheng et al., 2022; Bai et al., 2022; You et al., 2020).

**Layerwise Sparsity Ratios.** The distribution of sparsity over the layers of the network is shown to be a vital factor for sparse training procedures (Liu et al., 2023; Yin et al., 2023). The main approaches are:

- Uniform: each layer is assigned the sparsity ratio $S$, equal to the global sparsity ratio.

- Erdős–Rényi (ER): this approach (Mocanu et al., 2017) assigns higher sparsities to larger layers. A layer $l$ of size $n^{l-1} \times n^l$ receives a sparsity ratio that scales with
$$1 - \frac{n^{l-1} + n^l}{n^{l-1} \cdot n^l}.$$

- Erdős–Rényi-Kernel (ERK): this adaptation of ER is specifically designed by Evci et al. (2020) for convolutional layers, which consists of additional kernel dimensions. The calculation becomes
$$1 - \frac{n^{l-1} + n^l + w^l + h^l}{n^{l-1} \cdot n^l \cdot w^l \cdot h^l}.$$

In all experiments we allocate sparsity with ER; for convolutional layers we switch to its ERK variant. Some other sparse initialization methods involve loss function sensitivity initialization (Lee et al., 2018) or globally random allocations (Liu et al., 2023).

**Static Sparse Training.** Static methods train neural networks with a fixed sparse topology throughout the entire training process. While static sparse training requires fewer FLOPs compared to dense methods, it suffers from several fundamental limitations. The fixed topology prevents the network from adapting its structure during training, making the method highly sensitive to its initialization. This rigidity can create suboptimal paths for gradient flow and potentially limit the learning capacity of the network (Evci et al., 2022). There are promising directions to overcome these challenges (Nowak et al., 2024). Despite its limitations, static sparse training remains an important simple baseline in the sparse training field.

**Dynamic Sparse Training.** Methods in the domain of DST involve models that begin with sparse architectures and dynamically adapt their network topology during training. This process enables the network to explore alternative topologies in an evolutionary manner, gradually discovering more optimal network structures during training. SET (Mocanu et al., 2017), described in Section 2, has been successfully applied in different domains, from unsupervised and supervised learning (Nowak et al., 2023; Liu et al., 2021b; 2022; Yuan et al., 2021), to reinforcement learning (Grooten et al., 2023; Sokar et al., 2022) and continual learning (Yildirim et al., 2024). RigL (Evci et al., 2020) has also been widely adopted in research, having been applied in supervised learning (Nowak et al., 2023; Evci et al., 2022), reinforcement learning (Graesser et al., 2022; Tan et al., 2023), federated learning (Bibikar et al., 2022), and others. In a related line of work, Bellec et al. (2017) proposed DeepR, a similar approach where weights are pruned whenever the optimizer flips their sign.

# B    MODEL ARCHITECTURES

## B.1    RESNET

ResNet-50 is a 50-layer deep convolutional neural network that introduced the concept of residual learning to address the vanishing gradient problem in deep networks (He et al., 2016). It uses skip connections (or shortcuts) to bypass one or more layers, enabling the training of very deep networks by allowing gradients to flow directly through these connections. Its architecture consists of a series of residual blocks, each containing multiple convolutional layers and batch normalization layers.

Wide Residual Networks, such as Wide-ResNet28-10, are an extension of the original ResNets that focus on increasing the width (number of filters) of residual layers rather than their depth (Zagoruyko & Komodakis, 2016). This approach has been shown to improve performance while reducing computational complexity compared to very deep ResNets. Wide-ResNet achieves this by using fewer layers but with more convolutional filters per layer, which enhances feature learning and generalization.

## B.2    LLAMA

LLaMA-130M and LLaMA-350M are members of the decoder-only LLaMA family introduced by Touvron et al. (2023). The smaller variant comprises 12 transformer blocks with 768-dimensional hidden states and 12 attention heads, while the larger consists of 24 transformer blocks with 1024-dimensional hidden states and 16 attention heads. Both models retain the architectural choices of their larger counterparts, with LLaMA-130M fitting on commodity GPUs and achieving competitive perplexity for its size, serving as a strong lightweight backbone for further language-model experiments. LLaMA-350M offers greater capacity while maintaining the core architectural principles of the series. Our implementation of both models employs the open-source HuggingFace reproduction of LLaMA (Wolf et al., 2019).

## B.3    DQN

Our Atari agent uses a standard convolutional Deep Q-Network (DQN) (Mnih et al., 2013; 2015) as provided within CleanRL (Huang et al., 2022). The network takes a stack of 4 grayscale $84{\times}84$ frames and applies three conv blocks (channels 32, kernel $8 \times 8$, stride 4; 64, $4 \times 4$, 2; 64, $3 \times 3$, 1) with ReLU, followed by a flatten layer (3136 units) and two fully connected layers (1024 units), ending in a linear output of $|\mathcal{A}|$ Q-values. For ensembles, we instantiate either independent full networks or split the model after a shared convolutional backbone into $M$ heads.

## C DATASETS AND ENVIRONMENTS

### C.1 CIFAR-100

CIFAR-100 is an image classification dataset consisting of 60,000 color images sized 32×32 pixels, divided into 100 classes with 600 images per class. The dataset is split into 50,000 training and 10,000 test images (Krizhevsky, 2009).

**License**: CIFAR-100 is available for use in academic research. No official license was specified by the original authors.

### C.2 IMAGENET

ImageNet is a large-scale, high-resolution image database designed for research in visual object recognition. It contains over 14 million annotated images. The dataset has been foundational for advances in deep learning and computer vision, particularly through the ImageNet Large Scale Visual Recognition Challenge (ILSVRC), which includes 1,281,167 training images, 50,000 validation images, and 100,000 test images across 1,000 object categories (Deng et al., 2009).

**License**: ImageNet is available for non-commercial research and educational purposes under a custom non-commercial license. Access requires agreement to ImageNet terms of use, which restrict commercial exploitation.

### C.3 TINY-IMAGENET

Tiny-ImageNet is a subset of the full ImageNet dataset, containing 100,000 images of size 64×64 pixels, labeled across 200 classes. Each class has 500 training images, 50 validation images, and 50 test images, making it suitable for experiments requiring a smaller-scale version of ImageNet (Le & Yang, 2015; Deng et al., 2009).

**License**: Tiny-ImageNet is distributed for academic and research purposes only, under the same non-commercial terms as ImageNet.

### C.4 COLOSSAL CLEAN CRAWLED CORPUS (C4)

C4 is a large-scale text dataset constructed by cleaning and filtering web-crawled data from Common Crawl. It contains hundreds of gigabytes of English text, designed for training large language models and other NLP tasks. The dataset is filtered to remove low-quality and non-English content (Raffel et al., 2020).

**License**: The C4 dataset is used under the Open Data Commons Attribution License (ODC-By) v1.0, which allows free sharing, creation, and adaptation of the database provided proper attribution is maintained.

### C.5 ARCADE LEARNING ENVIRONMENT

The Arcade Learning Environment (ALE) provides a unified interface to dozens of Atari 2600 games for benchmarking reinforcement learning agents (Bellemare et al., 2013). In our experiments we use the `NoFrameskip-v4` gymnasium bindings with the standard DQN preprocessing within CleanRL Huang et al. (2022): no-op starts, action repeat (frameskip) of 4, grayscale, 84×84 resizing, and 4-frame stacking.

**License:** The ALE is available for academic research under the GNU General Public License.

## D NEUROTRAILS ALGORITHM

The NeuroTrails algorithm, detailed in Algorithm 1, aims to efficiently enhance the performance of neural network ensembles while significantly reducing their parameter footprint. The approach splits a given base architecture into a shared backbone and multiple sparse heads, initializing each part with a target sparsity ratio. Throughout training, each head independently processes shared representations from the backbone, enabling diverse predictions while leveraging common representation learning.

---

**Algorithm 1** NeuroTrails

---

1: **Input:** Base architecture $\mathcal{F}$, num. heads $M$, splitting point $\ell$, sparsity ratio $S$, drop fraction $p$.
2:
3: **Initialization Phase:**
4: Split $\mathcal{F}$ at block $\ell$ into shared blocks $\mathcal{F}_s$ and independent heads $\mathcal{F}_h$
5: Initialize $\mathcal{F}$ to sparsity ratio $S$
6:
7: **Training Phase:**
8: **for** each training iteration **do**
9:     $h_s = \mathcal{F}_s(x)$
10:     **for** each head $i \in \{1, \dots, M\}$ **do**
11:         $\hat{y}^i = \mathcal{F}_h^i(h_s)$
12:         $\mathcal{L}_i = \mathcal{L}(\hat{y}^i, y)$
13:     **end for**
14:     $\mathcal{L} = \frac{1}{M} \sum_{i=1}^{M} \mathcal{L}_i$
15:     $\theta_e \leftarrow \theta_e - \eta \nabla_{\theta_e} \mathcal{L}$
16:     **if** current iteration $\% \Delta T = 0$ **then**
17:         Prune $p$ fraction of parameters layerwise
18:         Grow $p$ fraction of parameters layerwise
19:     **end if**
20: **end for**
21:
22: **Inference Phase:**
23: Compute NeuroTrails prediction $\hat{y}$ by averaging the class probabilities predicted by each head $j$:

$$\hat{y} = \arg\max_i \left( \frac{1}{M} \sum_{j=1}^{M} \mathcal{F}_h^j(\mathcal{F}_s(x)) \right)_i$$

---

Periodically (every $\Delta T$ weight updates) the algorithm adjusts the network's topology through pruning and growing of parameters, controlled by a drop fraction $p$. During inference, predictions from individual heads are combined by averaging their output probabilities, resulting in a final aggregated prediction that leverages the strengths of each sparse pathway. See also Section 3 for further details.

# E TRAINING SETTINGS

## E.1 CODE REPOSITORIES

**Vision.** For computer vision experiments, we build upon the codebases from Liu et al. (2021b) and Dettmers & Zettlemoyer (2019), implementing our method throughout their existing sparse training library. The codebase from Dettmers & Zettlemoyer (2019) is released under the MIT license.

**Language.** For language experiments, we use the codebases from Li et al. (2025) and Zhao et al. (2024) as a foundation, implementing NeuroTrails in conjunction with the LLaMA architectures based on HuggingFace's reproduction (Wolf et al., 2019). The codebase from Zhao et al. (2024) is licensed under Apache 2.0.

**RL.** In the reinforcement learning setup we build upon the CleanRL codebase (Huang et al., 2022), adjusting the DQN architecture for NeuroTrails, TreeNet, and the Full Ensemble, as well as enabling independent training of these models. CleanRL is released under the MIT license.

## E.2 HYPERPARAMETERS

We describe the hyperparameters for our experiments. Tables 9 to 11 present the settings of our computer vision, language modeling and reinforcement learning experiments, respectively.

Table 9: Hyperparameters and settings for **computer vision** experiments.

| Parameter | Value |
|---|---|
| *Shared by all experiments* | |
|    optimizer | SGD with momentum |
|    learning rate schedule | 0.1× step decay at 25%, 50%, 75% of training |
|    ensemble aggregation | soft voting (mean of probabilities) |
| *CIFAR-100 and Tiny-ImageNet* (all baselines) | |
|    model | Wide-ResNet28-10 |
|    momentum | 0.9 |
|    initial learning rate | 0.1 |
|    batch size | 128 |
|    weight decay (L2) | $5 \cdot 10^{-4}$ |
|    training device CIFAR-100 | $1 \times$ NVIDIA V100 (16GB memory) |
|    training device Tiny-ImageNet | $4 \times$ NVIDIA A100 (40GB memory) |
|    approx. training time | 6.25h (CIFAR-100), 5.5h (Tiny-ImageNet) |
| *ImageNet* (all baselines) | |
|    model | ResNet-50 |
|    momentum | 0.875 |
|    initial learning rate | 0.256 |
|    batch size | 256 |
|    weight decay (L2) | $3.05 \cdot 10^{-5}$ |
|    training device | $4 \times$ NVIDIA A100 (40GB memory) |
|    approx. training time | 53h |
| *Static Sparse baseline* | |
|    sparsity ratio | varying (Section 5.1) |
|    sparsity initialization | ER |
|    topology update interval ($\Delta T$) | $\infty$ (no change) |
| *NeuroTrails* | |
|    sparsity ratio | varying (Section 4) |
|    sparsity initialization | ER |
|    DST drop fraction | $0.5 \cdot \text{cosine\_decay}(t)$ |
|    DST grow method | gradient (RigL) |
|    DST prune method | magnitude-based |
|    topology update interval ($\Delta T$) | 100 (CIFAR-100), 1000 (ImageNet) |
|    blocks in head | 8 (CIFAR-100), 10 (ImageNet) |
| *Pruning baseline* | |
|    sparsity ratio | varying (Section 5.1) |
|    pruning method | global magnitude |
|    pre-pruning phase | 250 epochs |
|    fine-tuning phase | 250 epochs |
| *TreeNet* | |
|    blocks in head | 8 (CIFAR-100), 10 (ImageNet) |
| *Full Ensemble* | |
|    training paradigm | Independent training |

**Computer Vision.** For CIFAR-100 and Tiny-ImageNet we train Wide-ResNet28-10 using SGD with momentum 0.9 and an initial learning rate of 0.1. The batch size is 128, the L2 regularization constant is set to 0.0005. For ImageNet, we follow the standard training regime (NVIDIA, 2024). We train ResNet-50 using SGD with a momentum of 0.875 and an initial learning rate of 0.256. The batch size is set to 256. We use L2 regularization with a fixed constant of 3.05e-05. For all computer vision datasets, the learning rate decreases by a factor of 10 after 25%, 50%, and 75% of training.

DST hyperparameters are set as follows: pruning and regrowing 50% of the available parameters at the beginning, with a cosine decay to 0 by the end of training. The $\Delta T$ topology update interval is set to 100 for CIFAR-100 and 1000 for ImageNet.

For the static sparse baseline of Section 5.1, we instantiate the model with the desired sparsity ratio at the start and subsequently train without adjusting the topology. For the pruning baseline, we employ global magnitude pruning to achieve the target sparsity ratio after the first 50% of training. After reaching the desired sparsity, we fine-tune the model for the remaining duration without making any further changes to its topology.

For ensemble training, we use the independent training paradigm, shown to work well by Jeffares et al. (2023), and train networks separately. At test time, we gather each network's predictions for the batch and average their raw probabilities, i.e., soft voting.

Table 10: Hyperparameters and settings for **language modeling** experiments.

| Parameter | Value |
|---|---|
| *Shared by all experiments* | |
| model size | 130M, 350M |
| optimizer | Adam |
| learning rate | $1.5e-3$ for 130M, $5e-4$ for 350M |
| learning rate schedule | cosine decay (min LR: 0.1×base) + warmup |
| learning rate warmup | 10% of update steps |
| weight decay | 0 (no decay) |
| ensemble aggregation | soft voting (mean of probabilities) |
| batch size | 512 |
| vocabulary size | 32,000 |
| max sequence length | 1024 |
| data type | bfloat16 |
| training device | $4 \times$ NVIDIA A100 (40GB memory) |
| approx. training time | 7.5h (130M), 40h (350M) |
| *NeuroTrails* | |
| sparsity ratio | 0.1 |
| sparsity initialization | ER with attention projections dense |
| DST drop fraction | $0.5 \cdot \text{cosine\_decay}(t)$ |
| DST grow method | gradient (RigL) |
| DST prune method | soft magnitude |
| soft magnitude temperature | 3.0 |
| topology update interval ($\Delta T$) | 50 steps |
| blocks in head | 8 (130M), 16 (350M) |
| *TreeNet* | |
| blocks in head | 8 (130M), 16 (350M) |
| *Full Ensemble* | |
| training paradigm | Independent training |

**Language Modeling.** For our language modeling experiments on the C4 corpus with LLaMA-130M/350M, we present the hyperparameters in Table 10. We train with Adam using a learning rate cosine-decay schedule (minimum LR set to 10% of the base) and a linear warmup over the first 10% of update steps. The batch size is 512 tokens, and we run on four A100 GPUs.

Table 11: Hyperparameters and settings for **reinforcement learning** experiments.

| Parameter | Value |
|---|---|
| *Shared by all experiments* | |
| optimizer | Adam |
| learning rate | $1e{-}4$ |
| discount $\gamma$ | 0.99 |
| batch size | 32 |
| replay buffer | 1,000,000 transitions |
| learning starts | 80,000 steps |
| train frequency | every 4 env steps |
| target net update period | every 1,000 env steps |
| target net update rate $\tau$ | 1.0 (hard copy) |
| $\epsilon$-greedy | linear decay $1.0 \rightarrow 0.01$ over $10\%$ of total steps |
| *NeuroTrails* | |
| sparsity ratio | 0.8 |
| sparsity initialization | ERK |
| DST drop fraction | $0.5 \cdot \text{cosine\_decay}(t)$ |
| DST grow method | random (SET) |
| DST prune method | magnitude |
| topology update interval ($\Delta T$) | 2000 grad steps |
| blocks in head | 4 |
| *TreeNet* | |
| blocks in head | 4 |

Our NeuroTrails models employ dynamic sparse training: at every $\Delta T = 50$ steps we drop and regrow a fraction $p$ of the weights, decaying the drop fraction to zero by the end of training. We allocate 8 or 16 transformer blocks per head, matching the approximate ⅓ shared-backbone setting used in our vision experiments. This configuration lets each head discover and adapt its own topology while respecting the overall FLOP budget.

As a reference, we split the TreeNet baseline likewise into a shared backbone and 8 or 16-block heads. For the dense-ensemble baseline, we train three independent LLaMA-130M models from scratch under the same schedule and aggregate their outputs via the same soft-voting scheme.

Experiments on C4, ImageNet, and Tiny-ImageNet have been carried out using distributed training on 4 NVIDIA A100 GPUs, while for CIFAR-100 training was done on a single NVIDIA V100.

**Reinforcement Learning.** We train for 10M steps (40M frames) on the Atari environments: `Asterix`, `BeamRider`, `Breakout`, `Seaquest`, `SpaceInvaders`, and `UpNDown`. Every 100K steps we evaluate the model on 10 episodes. We average these evaluation returns over the last 10% of training (100 eval episodes in total) for more reliable results, following (Graesser et al., 2022; Grooten et al., 2023). We take the interquartile mean to be more robust against outliers, as proposed by Agarwal et al. (2021).We train with standard Atari preprocessing (No-op starts, frameskip 4, reward clipping) as is default in CleanRL (Huang et al., 2022). Hyperparameters are presented in Table 11. As mentioned in Section 4.3, we train the heads independently, but sample actions in the environment jointly as we aim to collect the best data. We find that training independently is crucial, but joint or independent sampling does not make a large difference. For NeuroTrails and TreeNet we split the backbone after the three convolutional layers (see Appendix B). NeuroTrails adjusts the sparse topology after every $\Delta T = 2000$ weight updates, with an initial drop fraction $p = 0.5$ annealed through cosine decay. We simply use magnitude pruning and random growth, as SET (Mocanu et al., 2017) has been shown to work just as well as RigL (Evci et al., 2020) in RL (Graesser et al., 2022).

### E.3 TRAINING SCHEDULES

Building on the observation that dense models tend to overfit once training is prolonged, whereas sparse networks keep improving as they are still refining both weights and topology (Liu et al.,

2021b), we follow the recipe of Evci et al. (2020) and extend the training of sparse variants by at most $1/(1-S)$, so that its total FLOPs never exceeds that of the dense counterpart. Exact training schedules appear in Tables 12 to 15.

Table 12: Training cost comparison on CIFAR-100 (Wide-ResNet28-10). Baselines marked with * are from Havasi et al. (2021), ** from Liu et al. (2022), and *** from Lee & Lee (2024).

| Method | Train Epochs | Train FLOPs ($\downarrow$) |
|---|---|---|
| Single Dense * | 250 | 3.6e17 |
| MIMO ($M = 3$) * | 250 | 1.00× |
| EDST Ensemble ($M = 7$) ($S = 0.9$) ** | 850 | 0.57× |
| DST Ensemble ($M = 3$) ($S = 0.8$) ** | 3×250 | 1.01× |
| Batch Ensemble ($M = 4$) * | 250 | 1.10× |
| NFE ($M = 3$) *** | 200 | 1.02× |
| TreeNet ($M = 3$) | 250 | 2.91× |
| Full Ensemble ($M = 3$) | 3×250 | 3.00× |
| NeuroTrails ($M = 3$) ($S = 0.8$) | 450 | 0.85× |
| NeuroTrails ($M = 5$) ($S = 0.9$) | 450 | 0.67× |

In Table 12 we report the number of epochs and relative training FLOPs on CIFAR-100 with Wide-ResNet28-10. The dense model runs for 250 epochs (3.6e17 FLOPs). Sparse methods such as EDST ($S = 0.9$) and DST ($S = 0.8$) run up to 850 or 750 total epochs—to compensate for their reduced per-epoch cost, while other baselines (MIMO, BatchEnsemble, NFE, TreeNet) stay close to 250. NeuroTrails is trained for 450 epochs, calibrated to maintain computational efficiency significantly below a single dense network, as measured by total training FLOPs.

Table 13 shows the analogous schedule on ImageNet (ResNet-50). The single-model baseline uses 90 epochs (4.8e18 FLOPs); EDST and DST extend to 310 and 400 total epochs respectively, whereas NeuroTrails ($S = 0.7$) runs for 270 epochs—staying close to a single dense model's compute budget. Notably, most baselines tune their schedules so that total training FLOPs remain close to the 1.0× dense reference. We recognize that reported training lengths vary substantially across papers; to ensure fidelity to each comparison, we simply report each method's published schedule when taking values from the original works. We encourage the ensembling field to always publish the full training schedules of all baselines and adopt more consistent training protocols, to enable clearer comparisons.

Table 13: Training cost comparison on ImageNet (ResNet-50). Baselines marked with * are from Havasi et al. (2021), ** from Liu et al. (2022).

| Method | Train Epochs | Train FLOPs ($\downarrow$) |
|---|---|---|
| Single Dense * | 90 | 4.8e18 |
| MIMO ($M = 2$) ($\rho = 0.6$) * | 150 | 1.00× |
| EDST Ensemble ($M = 4$) ($S = 0.8$) ** | 310 | 0.87× |
| DST Ensemble ($M = 2$) ($S = 0.8$) ** | 2×200 | 1.12× |
| Batch Ensemble ($M = 4$) * | 4×135 | 1.10× |
| TreeNet ($M = 3$) | 180 | 2.91× |
| Full Ensemble ($M = 4$) * | 4×90 | 4.00× |
| NeuroTrails ($M = 3$) ($S = 0.7$) | 270 | 1.10× |

For C4 pretraining with LLaMA models, Table 14 compares the number of weight-update steps, total tokens seen, and training FLOPs across different methods. The dense baseline for LLaMA-130M performs 10,000 updates (1.0B tokens), while NeuroTrails ($S = 0.1$) scales to 11,111 steps (1.1B tokens), precisely matching TreeNet's training FLOPs via the $1/(1-S)$ rule. A similar scaling applies to the larger LLaMA-350M model, where NeuroTrails with sparsity 0.1 takes 44,444 steps (4.4B tokens), again equating to the training FLOPs of the dense counterpart.

Table 14: Compute comparison on C4 pretraining (LLaMA-130M/350M).

| Method | Train Updates | Tokens seen | Training FLOPs ($\downarrow$) |
|---|---|---|---|
| *LLaMA-130M* | | | |
| Single Dense | 10,000 | 1.0B | 3.5e18 |
| TreeNet ($M = 3$) | 10,000 | 1.0B | 2.21× |
| Full Ensemble ($M = 3$) | 3×10,000 | 1.0B | 3.00× |
| NeuroTrails ($M = 3$) ($S = 0.1$) | 11,111 | 1.1B | 2.21× |
| *LLaMA-350M* | | | |
| Single Dense | 40,000 | 4.0B | 3.5e18 |
| TreeNet ($M = 3$) | 40,000 | 4.0B | 2.27× |
| Full Ensemble ($M = 3$) | 3×40,000 | 4.0B | 3.00× |
| NeuroTrails ($M = 3$) ($S = 0.1$) | 44,444 | 4.4B | 2.27× |

Finally, Table 15 gives epochs and FLOPs on Tiny-ImageNet (see results in Appendix G). The baseline is 100 epochs (3.2 e17 FLOPs), NFE variants pretrained on CIFAR-100 add a 200-epoch warm-up, and NeuroTrails heads ($S = 0.8, 0.9$) run 200 epochs—staying well below the $1/(1-S)$ rule—achieving 0.74× training FLOPs compared to a single dense model.

Table 15: Training cost comparison on Tiny-ImageNet (Wide-ResNet28-10). Baselines marked with * are from Lee & Lee (2024).

| Method | Train Epochs | Train FLOPs ($\downarrow$) |
|---|---|---|
| Single Dense | 100 | 3.2e17 |
| NFE ($M = 2$) ($S = 0.25$) (pretrained on CIFAR-100) * | 100(+200 pre) | 0.76× |
| NFE ($M = 3$) ($S = 0$) (pretrained on CIFAR-100) * | 100(+200 pre) | 1.01× |
| TreeNet ($M = 3$) | 100 | 2.91× |
| Full Ensemble ($M = 3$) | 3×100 | 3.00× |
| NeuroTrails ($M = 3$) ($S = 0.8$) | 200 | 0.94× |
| NeuroTrails ($M = 5$) ($S = 0.9$) | 200 | 0.74× |

# F  METRICS

**Test Accuracy** quantifies the model's generalization capability by measuring the proportion of correctly classified samples in a held-out test set (Hastie et al., 2001). Higher test accuracy indicates better classification performance. This fundamental metric serves as the primary indicator of classification quality, though it should be interpreted in conjunction with uncertainty-aware metrics.

**Negative Log-Likelihood** (NLL) (Hastie et al., 2001) evaluates the quality of probabilistic predictions by computing the negative logarithm of predicted probability assigned to the true class, where lower values indicate superior uncertainty estimation.

**Expected Calibration Error** (ECE) (Guo et al., 2017; Naeini et al., 2015) measures the model calibration by calculating the discrepancy between the prediction confidence and the empirical accuracy between different confidence bins. Lower ECE indicates better calibration, meaning the model confidence estimates align more closely with actual accuracy. We used 15 bins to estimate this metric, following (Guo et al., 2017).

**Perplexity** is a statistical measure used to evaluate how well a probabilistic model predicts a sample, commonly applied in natural language processing to assess language models (Jelinek et al., 1977). It quantifies the model's uncertainty when predicting the next token in a sequence by calculating the exponential of the average negative log-likelihood, with lower perplexity values indicating higher quality models.

**Prediction Disagreement** quantifies the extent to which multiple models produce differing outputs for the same input (Skalak, 1996). Higher disagreement often indicates areas of uncertainty, offering insight into decision boundaries and aiding in the detection of out-of-distribution samples.

**Throughput** for a machine learning model is defined as the number of data samples processed by the model per unit time, typically measured in items or images per second.

**Latency** for a machine learning model is the time taken for the model to process a data sample from input to output, usually measured in milliseconds or seconds per batch.

**FLOPs** refers to the number of floating-point operations a model performs during training or inference. It serves as a measure of the model's computational complexity and efficiency. We adopt the FLOPs calculation methodology from Evci et al. (2020): For a given dense architecture with forward pass FLOPs $f_D$ and a sparse version with FLOPs $f_S$, the total FLOPs required for one update step scale with $3 \cdot f_S$ and $3 \cdot f_D$ FLOPs, respectively. This scaling arises because training consists of two main steps: (1) a forward pass, where activations are computed and stored layer-by-layer to evaluate the loss, and (2) a backward pass, where the error signal is back-propagated to compute gradients. The backward pass is approximately twice as expensive as the forward pass, as each layer must compute gradients with respect to both its parameters *and* its input activations. For further details we refer to Appendix H of the RigL paper (Evci et al., 2020).

**Interquartile Mean** (IQM) is a robust summary statistic that averages only the middle 50% of scores, i.e., after trimming the lowest and highest quartiles, and is recommended for RL reporting due to its reduced sensitivity to outliers and occasional run failures (Agarwal et al., 2021). We report IQM with the standard error of the mean (s.e.m.), where s.e.m. is computed over the retained middle 50% only.

## G    ADDITIONAL RESULTS

### G.1    PARAMETER-MATCHED LANGUAGE MODEL STUDY

To isolate the effect of *sparsity + multi-heads* from sheer model size, we build a parameter-matched NeuroTrails variant whose total parameter budget is essentially identical to that of a larger dense model. We compare a single dense LLaMA-250M model with a NeuroTrails variant of LLaMA-130M which we designed to match the number of parameters. It uses 3 heads, with just 7 blocks per head (instead of our default 8), and a sparsity ratio of 13%. We train both for exactly the same number of update steps (10k), meaning both models see approximately 1B training tokens. As shown in Table 16, NeuroTrails delivers a modestly lower validation perplexity (26.48 vs. 26.59), despite having slightly fewer parameters than the 250M dense baseline.

Table 16: Size–efficiency comparison: a 250M-parameter single dense model versus a NeuroTrails variant with essentially the same parameter budget.

| Method | Perplexity ($\downarrow$) | Parameters ($\downarrow$) | Train FLOPs ($\downarrow$) | Infer. FLOPs ($\downarrow$) |
|---|---|---|---|---|
| Single Dense | 26.59 | 247.37M | 7.0e18 | 4.56e11 |
| NeuroTrails ($M = 3, S = 0.13$) | **26.48** | 245.66M | 1.0× | 1.0× |

### G.2    TINY-IMAGENET

In this section, we present results from training our model on the Tiny-ImageNet benchmark in Table 17. NeuroTrails outperforms the baselines trained from scratch at a competitive accuracy. Furthermore, NeuroTrails requires significantly fewer inference FLOPs than any other model, using only 34% of the budget of a single dense model.

Table 17: Performance on Tiny-ImageNet (Wide-ResNet28-10). NeuroTrails and TreeNet have 8 blocks in the heads, with 4 in the backbone. Results marked with * are from Lee & Lee (2024), who did not report NLL and ECE, and used a pretrained model instead of training from scratch.

| Method | Accuracy ($\uparrow$) | NLL ($\downarrow$) | ECE ($\downarrow$) | Train FLOPs ($\downarrow$) | Infer. FLOPs ($\downarrow$) |
|---|---|---|---|---|---|
| Single Dense | 66.5 | 1.510 | 0.121 | 3.2e17 | 10.5e9 |
| NFE ($M$=2, $S$=0.25) (pretrained on CIFAR-100) * | **71.0** | - | - | 0.76× | 0.76× |
| NFE ($M$=3, $S$=0) (pretrained on CIFAR-100) * | 70.6 | - | - | 1.01× | 1.01× |
| TreeNet ($M = 3$) | 69.6 | 1.310 | 0.118 | 2.91× | 2.91× |
| Full Ensemble ($M = 3$) | 70.8 | 1.273 | 0.115 | 3.00× | 3.00× |
| NeuroTrails ($M = 3$) ($S = 0.8$) | 70.7 | 1.322 | 0.117 | 0.94× | 0.47× |
| NeuroTrails ($M = 5$) ($S = 0.9$) | 70.9 | 1.251 | 0.115 | **0.74×** | **0.34×** |

# H  SPARSITY RATIO ANALYSIS

This section examines the relationship between sparsity ratios and the accuracy of the model. To ensure a controlled analysis, we fix the ensemble size at 3, set the backbone sharing factor to 8, and vary the sparsity ratio from fully dense, which corresponds to being 0% sparse, to 99% sparse.

Our experimental results indicate that for CIFAR-100, optimal performance is achieved at 80% sparsity, where the model retains only 20% of its original parameters (Figure 8). Interestingly, the dense model performs worse, probably due to the overparameterization introduced during the ensemblification process, leading to overfitting. These results underscore the critical role of sparsity as a regularization mechanism in NeuroTrails, which enhances the model's predictive performance.

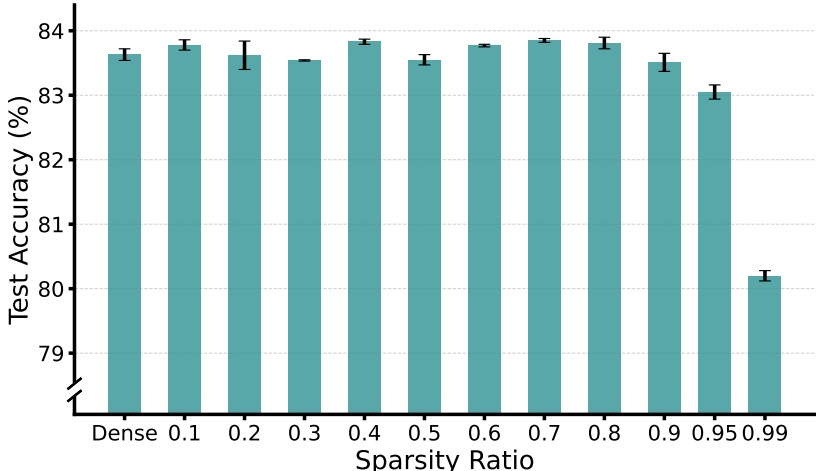

Figure 8: Impact of sparsity ratio on accuracy in a NeuroTrails model with three heads ($M = 3$, Wide-ResNet28-10) on CIFAR-100. The 80% sparse configuration emerges as the optimal choice, though the performance differences across closest competitors are notably small.

## H.1  ULTRA SPARSE

We provide additional experiments with ultra-sparse networks in Table 18. While performance gradually declines with increasing sparsity, models still maintain strong accuracy even in the ultra-sparse regime. These regimes are crucial for real-world deployment, especially on devices with limited computational capacity. We present examples of real-time inference gains in Section 5.4 and Appendix I.

Table 18: NeuroTrails performance in ultra-sparse regimes on CIFAR-100.

| Model on CIFAR-100 | Accuracy ↑ (%) |
|---|---|
| NeuroTrails ($M = 3$) ($S = 0.95$) | $83.05 \pm 0.11$ |
| NeuroTrails ($M = 3$) ($S = 0.99$) | $80.20 \pm 0.08$ |
| NeuroTrails ($M = 5$) ($S = 0.95$) | $83.48 \pm 0.04$ |
| NeuroTrails ($M = 5$) ($S = 0.99$) | $81.06 \pm 0.15$ |
| NeuroTrails ($M = 5$) ($S = 0.995$) | $79.24 \pm 0.17$ |

# I  REAL-TIME INFERENCE GAIN

In this section we examine the discrepancy between theoretical gains from sparsity and the practical speedups achieved on existing hardware. Despite growing interest in sparsity-aware computation, current hardware support remains limited. Notable developments include the DeepSparse library (NeuralMagic, 2021), which offers CPU-level sparse acceleration through an accessible Python library,

and dedicated hardware solutions like Cerebras chips (Cerebras, 2024). However, deploying models on Cerebras hardware typically requires proprietary access, which restricts broader experimentation. By contrast, DeepSparse provides an immediate, open solution for evaluating sparse inference performance. There are multiple other works in the direction of truly sparse implementations on GPU hardware (Schultheis & Babbar, 2023; Liu et al., 2021a; Curci et al., 2021; Wesselink et al., 2024).

As illustrated in Figure 9, the theoretical number of sparse FLOPs (shown in blue) decreases substantially with increasing sparsity, dropping well below the latency of a single dense model (indicated by the dashed line) at 80% sparsity model with 3 heads. However, the extent to which these theoretical savings translate into real-world latency reductions is highly dependent on hardware capabilities. In the absence of dedicated sparsity acceleration (yellow), inference latency remains constant across sparsity ratios. Partial hardware support through DeepSparse integration (orange), on the other hand, enables meaningful efficiency gains—particularly at ultra-high sparsity ratios (e.g., 95–99%). These findings highlight the promise of sparse model execution under current constraints and underscore the need for further research into hardware architectures optimized for sparsity.

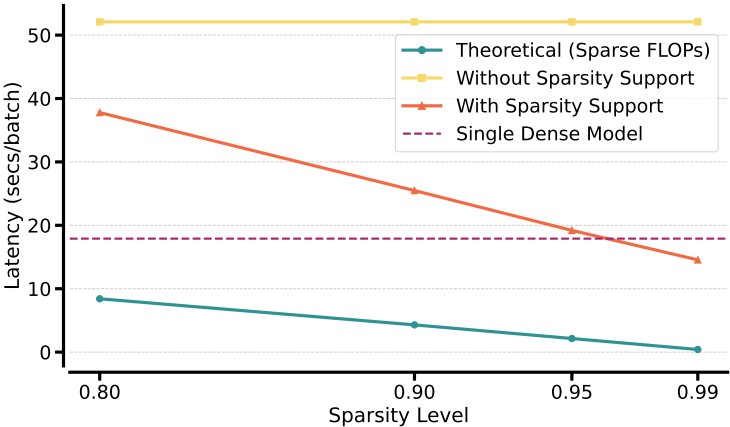

Figure 9: Inference latency comparison across NeuroTrails models with increasing sparsity ratios ($M = 3$), using DeepSparse for sparse inference acceleration.

## J  GOLDILOCKS IMAGE SAMPLES

We defined *prediction conflict* in Section 5.2 and present examples in this section that illustrate this phenomenon. In short, we theorize that models with high prediction diversity among ensemble members may suffer from aggregation inefficiency when these predictions conflict with each other. For the sake of conciseness, we refer to model with 8 blocks in head as NeuroTrails 8, and model with 12 blocks in head as NeuroTrails 12 in this section.

This phenomenon is demonstrated in Figure 10, where we observe that the lower prediction disagreement in NeuroTrails 8 consistently produces better prediction estimates, while NeuroTrails 12 exhibits signs of aggregation breakdown, resulting in erroneous predictions. For example, NeuroTrails 8 predicts poppy–poppy–worm for the first image, whereas NeuroTrails 12 predicts orange–sunflower–poppy. The ground truth label is poppy, making the former a correct prediction and the latter an incorrect one. In this instance, the higher prediction diversity in NeuroTrails 12 results in conflicting outputs that hinder accurate aggregation. This illustrates how excessive diversity among predictors can degrade ensemble performance, supporting our hypothesis that prediction conflict undermines aggregation efficiency.

It is important to note that both models achieve high accuracy, with minimal differences between them (83.81% for NeuroTrails 8 versus 83.59% for NeuroTrails 12). While our proposed explanation of prediction conflict may account for this difference, we acknowledge that alternative factors could also contribute to these observations.

NeuroTrails (8 blocks) predictions:
poppy, poppy, worm
Aggregate: poppy

NeuroTrails (12 blocks) predictions:
orange, sunflower, poppy
Aggregate : sunflower

True: poppy

NeuroTrails (8 blocks) predictions:
forest, forest, forest
Aggregate: forest

NeuroTrails (12 blocks) predictions:
plain, willow_tree, forest
Aggregate : plain

True: forest

NeuroTrails (8 blocks) predictions:
skyscraper, skyscraper, skyscraper
Aggregate: skyscraper

NeuroTrails (12 blocks) predictions:
bridge, skyscraper, skyscraper
Aggregate : bridge

True: skyscraper

NeuroTrails (8 blocks) predictions:
bear, bear, elephant
Aggregate: bear

NeuroTrails (12 blocks) predictions:
elephant, seal, bear
Aggregate : elephant

True: bear

Figure 10: Direct prediction comparison between NeuroTrails models with 8 and 12 blocks in their heads. NeuroTrails-8 exhibits a prediction diversity level that is *just right*, enabling it to produce more accurate results than its counterpart.

