# OpenReview forum: "Training with Dynamic Sparse Heads as the Key to Effective Ensembling"
_ICLR.cc/2026/Conference — ICLR 2026 Conference Desk Rejected Submission_

### Official Review · Reviewer_YJp6 · 2025-10-28

**Soundness:** 3
**Presentation:** 4
**Contribution:** 1
**Rating:** 2
**Confidence:** 3

**Summary:**

The authors introduce an efficient training and inference methodology, called NeuroTrails. Neurotrails starts by pre-training a base model. It then initializes multiple heads, as in TreeNet. Weights are sparsified. Then, during training, weights can be reactivated using RigL. This results in a highliy performant network with few training and inference FLOPS.

**Strengths:**

* The paper shows strong predictive performance on a large number of tasks.
* The paper is very clearly written.
*  Ablations are interesting.

**Weaknesses:**

* The single biggest weakness is a practical one. In Figure 7, the authors show a pareto frontier wrt throughput . But this is just on CPU. In practice, GPU's have become ubiquitous. How is speedup on GPUs with current sparse librarries? I suspect it is very difficult to get competitive walltime speedup, compared to methods which use dense multiplications (like MIMO). I believe this severly limits the utility of this method, and a solution does not appear to be on the horizon.

* The novelty feels weak: It's an combination of two known methods:. TreeNet and RigL. Both methods have existed for a long time.

* The most extensive comparisons are on CIFAR-100 and ImageNet. Of these two, only ImageNet represents a difficult and interesting problem.

**Questions:**

* How do results compare with using lower precision weights instead? [1]
* How practical are NeuroTrails on GPUs?
* How do baselines perform on language modelling and RL tasks?
[1] Zhou, S., Wu, Y., Ni, Z., Zhou, X., Wen, H., and Zou, Y. Dorefa-net: Training low bitwidth convolutional neural networks with low bitwidth gradients. arXiv preprint arXiv:1606.06160, 2016.

---

> ### Author Response · Authors · 2025-11-28
>
> Dear reviewer YJp6,
>
> We appreciate that you’ve found our paper to be “very clearly written”, with interesting ablations and strong performance on a large number of tasks.
>
> Addressing your points and questions:
>
> W1-Q2. We believe that the achieved throughput gains on CPU are already greatly useful in widespread devices such as smartphones, where AI is used a lot nowadays. To achieve further speedups on GPUs as well, we implemented Structured RigL (SRigL) [1] into NeuroTrails, which is able to use the N:M semi-structured sparsity functionality of modern GPUs. The results show that it performs almost as well as RigL in terms of eval perplexity, while having potential to improve inference efficiency.
>
> |   Method      | Perplexity  (↓)  |
> |:-----------------|:---------------|
> |    NeuroTrails (M=3, S=0.1, RigL)    |   26.00 |
> |    NeuroTrails (M=3, S=0.1, SRigL)  |    26.27  |
>
> The inference speed on GPUs when measured simply as-is, provides the following with batch size 128.
>
> | Method          | Latency (ms)    | Throughput (samples/sec) |
> | :----------------- | :------------------- | :----------------------- |
> | Single Dense  | 20.66 ± 0.07   | 6195.73                  |
> | TreeNet          | 44.82 ± 0.12   | 2855.66                  |
> | NeuroTrails     |  44.88 ± 0.12  |  2852.25                  |
> | Full Ensemble | 62.04 ± 0.20   | 2063.09                 |
>
> We see that NeuroTrails is just as fast as TreeNet, significantly quicker than the Full Ensemble. With access to true N:M semi-structured sparsity support NeuroTrails’ speed could be improved even further, while outperforming all baselines in terms of accuracy, perplexity, and reward.
>
> W2. To the best of our knowledge, we are the first to thoroughly investigate the use of dynamic sparse training in the ensembling approach of TreeNet, especially across multiple fields (computer vision, language modeling, reinforcement learning). Furthermore, we present deep empirical insights into the Goldilocks zone of prediction diversity that the splitting point $\ell$ provides, which was described as a “valuable and insightful contribution to the field of ensemble learning” by reviewer oJ41.
>
> W3-Q3. We were able to compare the most baselines in the field of computer vision, because that is where most of the ensembling literature has performed experiments in previous years. We are the first to extensively analyze the performance of NeuroTrails and similar baselines in other fields, demonstrating strong results in language modeling and reinforcement learning as well.
>
> Q1. As described in Appendix E.2, we already train our language models in bfloat16, while our computer vision and reinforcement learning experiments were performed in float32 precision. The results across Tables 2–6 demonstrate that NeuroTrails works well in both settings.
>
> Thank you for your helpful comments and for reviewing our work. We hope that our responses alleviate your concerns and would be grateful if you could kindly reconsider your score.
>
> [1] https://arxiv.org/abs/2305.02299

---

### Official Review · Reviewer_er7G · 2025-10-29

**Soundness:** 2
**Presentation:** 3
**Contribution:** 2
**Rating:** 4
**Confidence:** 3

**Summary:**

This paper proposes NeuroTrails, a model-agnostic ensembling paradigm that shares an early backbone across multiple dynamically sparse heads, trained with dynamic sparse training (DST) and periodically re-wiring masks to explore diverse “neural trails.” At training time, each head receives its own loss and the losses are averaged; at inference time, logits are averaged while the shared backbone is computed once. The key claim is that, by selecting an appropriate split point and maintaining sparsity, NeuroTrails achieves a “Goldilocks” level of prediction diversity that improves accuracy/robustness while cutting inference FLOPs versus full ensembles.

**Strengths:**

1) Clear architectural idea. The split-backbone multi-head design is simple and broadly applicable.
2) Diverse empirical coverage. Results span supervised vision, language pretraining and RL, with gains over single models, TreeNet, MIMO/BatchEnsemble, and full ensembles.
3) Insight on ensemble diversity. The “PD Goldilocks zone” is a useful observation: the best CIFAR-100 result has lower disagreement than over-separated heads or a full ensemble, suggesting an optimal diversity band rather than “more is better.”

**Weaknesses:**

1) Novelty relative to TreeNet+DST is incremental. The core ingredients of the proposed method are shared early trunk (TreeNet) plus sparse training (RigL/SET) and multi-head branching, which are known individually. The paper should better isolate what new emerges from their combination beyond “do both”.
2) Fairness/compute accounting needs tightening. The paper states sparse variants are trained longer (up to a factor ~1/(1−S)) while “keeping training FLOPs below dense counterparts,” but it is unclear whether all comparison baselines are matched on total training FLOPs or wall-clock time. It lacks a single, consolidated compute budget table to confirm matching across methods for each domain.
3) Practicality on GPUs remains uncertain. Reported speedups rely on CPU(DeepSparse) and discuss emerging hardware. There is no measured GPU latency/throughput for standard frameworks where unstructured sparsity often yields limited wins. Add end-to-end latency on commodity GPUs, or show an engineering pathway to on-GPU speedups at target sparsities.
4) Diversity measurement is narrow. PD is informative but coarse. Add CKA/representation similarity, error correlation, per-class co-disagreement, and calibration change from ensembling to establish the “Goldilocks” claim more rigorously (and check that higher PD does not merely correlate with miscalibration).
5) Language experiments are small-scale. LLaMA-130M/350M are instructive but far from modern LLM scales. It is unclear if gains persist at ≥1–7B or with instruction-tuned models.
6) Outdated comparison baselines (mostly ≤2022). The comparison set skews toward older methods , omitting strong recent contenders from 2023–2025.

**Questions:**

1) For each table, are total training FLOPs and wall-clock matched across all baselines, including dense/full ensembles?
2) What ∆T, prune fraction p, regrowth rule, and layerwise sparsity targets were used per domain? How sensitive are results to these hyperparameters?
3) Can you report end-to-end inference latency/throughput on common GPUs for ImageNet and CIFAR-100 models at differend S values and compare with the dense model?
4) Do accuracy gains persist if PD is held constant (e.g., via temperature/logit scaling or noise)? Can you add CKA/error-correlation analyses to substantiate the Goldilocks hypothesis?
5) Have you tested ≥1B-parameter LLM backbones? Are attention projections the only dense components? If so, what is the impact of sparsifying them?
6) Can you add a baseline that is TreeNet with DST but without shared mask, to isolate the specific gain from head-wise DST.

---

> ### Author Response · Authors · 2025-11-28
>
> Dear reviewer er7G,
>
> Thank you for finding our method to be “broadly applicable”. We appreciate that you’ve found the “diverse empirical coverage” and “insight on ensemble diversity” to be among our work’s main strengths.
>
> Addressing your points and questions:
>
> W1. To the best of our knowledge, this is the first work to thoroughly study dynamic sparse training within the TreeNet ensemble framework, especially spanning multiple fields (computer vision, language modeling, and reinforcement learning). In addition, we provide extensive empirical analysis of the Goldilocks zone of prediction diversity induced by the splitting layer $\ell$, which you mentioned as one of our strengths.
>
> W2. We describe the intricacies of the training schedules used by all baselines in detail in Appendix E.3, including the consolidated compute budget table you’re looking for. We agree that it is not ideal that training lengths can vary substantially across papers. To ensure fidelity to each comparison, we report each method’s published schedule when taking values from the original works. As stated in our Appendix E.3: “We encourage the ensembling field to always publish the full training schedules of all baselines and adopt more consistent training protocols, to enable clearer comparisons.”
>
> W3. The achieved CPU throughput gains are already very practical for common devices like smartphones, where AI workloads are increasingly prevalent. To additionally obtain speedups on GPUs, we incorporated Structured RigL (SRigL) [1] into NeuroTrails, enabling the use of N:M semi-structured sparsity on modern GPUs. The results show that it performs almost as well as RigL in terms of eval perplexity, while having potential to improve inference efficiency.
>
> |   Method      | Perplexity  (↓)  |
> |:-----------------|:---------------|
> |    NeuroTrails (M=3, S=0.1, RigL)    |   26.00 |
> |    NeuroTrails (M=3, S=0.1, SRigL)  |    26.27  |
>
> The inference speed on GPUs when measured simply as-is, provides the following with batch size 128.
>
> | Method          | Latency (ms)    | Throughput (samples/sec) |
> | :----------------- | :------------------- | :----------------------- |
> | Single Dense  | 20.66 ± 0.07   | 6195.73                  |
> | TreeNet          | 44.82 ± 0.12   | 2855.66                  |
> | NeuroTrails     |  44.88 ± 0.12  |  2852.25                  |
> | Full Ensemble | 62.04 ± 0.20   | 2063.09                 |
>
> We see that NeuroTrails is just as fast as TreeNet, significantly quicker than the Full Ensemble. With access to true N:M semi-structured sparsity support NeuroTrails’ speed could be improved even further, while outperforming all baselines in terms of accuracy, perplexity, and reward.
>
> W4. We agree that prediction disagreement (PD) is only one lens on ensemble diversity. We chose PD because it directly measures disagreement at the prediction level and is standard in the ensemble-learning literature. Importantly, our “Goldilocks” observation is not based on PD in isolation: for each choice of splitting layer $\ell$, we jointly report PD and task performance, and consistently find that intermediate PD yields the best results. This non-monotonic relationship argues against the trivial explanation that “higher PD merely reflects miscalibration.” We fully agree that complementary measures would provide an even richer picture, but we leave these for follow-up work. Please do see our additional analysis in response to reviewer oJ41, where we investigate the sensitivity of hyperparameters such as the sparsity ratio and the choice of DST method on the diversity of NeuroTrails.
>
> W5. Regarding language-scale, our goal in this work is to study NeuroTrails in a regime that is still realistic for academic labs while already non-trivial in size. The LLaMA-130M/350M backbones correspond, in full ensembles with M=3, to approximately 390M and 1.05B parameters respectively, i.e., already in the “~1B-class’’ range. Running multiple 1–7B models per configuration would require substantially larger compute than is available to us and is beyond the scope of this paper. Crucially, the proposed mechanism (a shared trunk with dynamically sparse, diversified heads) is architecture-agnostic and does not rely on any property specific to small or mid-sized models.

---

> > ### Author Response · Authors · 2025-11-28
> > **Continuation of Previous Response**
> >
> > W6. We agree that adding more recent ensemble baselines from 2023–2025 would further strengthen the empirical study. Our current baseline set was driven by two constraints: (i) we focus on methods that are both widely cited and representative of the main ensembling paradigms (independent full ensembles, TreeNet-style branching, and shared-parameter approaches such as MIMO/BatchEnsemble), and (ii) for which public implementations and sufficiently detailed training schedules are available, so that we can meaningfully integrate dynamic sparse training while matching compute (see Appendix E.3). Many recent variants either target very specific application regimes or do not provide enough training details to allow a fair, compute-matched comparison. In the revised version, we will clarify these selection criteria and expand the related-work discussion to cover recent ensemble and sparse-LLM works from 2023–2025, emphasizing that NeuroTrails is complementary and can in principle be combined with such methods.
> >
> > Q1. We provide the training schedule details in Appendix E.3, including the total FLOPs used by each baseline.
> >
> > Q2. In Appendix E.2 you’ll find our hyperparameters as set per domain (computer vision in Table 9, language modeling in Table 10, and reinforcement learning in Table 11). The sensitivity to hyperparameters such as the backbone length (Section 5.1), ensemble size (Section 5.3), and sparsity ratio (Appendix H) are thoroughly studied in our work.
> >
> > Q3. Thank you for the suggestion, we ran additional experiments to address this. Please see our response at W3, which discusses the inference speed of NeuroTrails on GPUs, even with Structured RigL.
> >
> > Q4. We agree that PD is only one way to probe ensemble diversity. In this work, our “Goldilocks” observation is based on how accuracy co-varies with PD as we move the splitting layer $\ell$, i.e., using the naturally induced changes in PD rather than explicitly constraining PD via temperature scaling or injected noise.
> >
> > Q5. Our largest language instantiations use LLaMA-130M/350M backbones, which in full ensembles with $M=3$ correspond to approximately 390M and 1.05B parameters respectively, already pushing our available compute. In the current transformer setup, we apply DST to the large linear layers (e.g., feed-forward/MLP blocks) while indeed only keeping the key components, attention projections, dense. This follows common practice in sparse-transformer work and avoids stability issues from aggressively sparsifying relatively small but structurally important sub-blocks.
> >
> > Q6. We already run NeuroTrails without shared masks across the heads. Each head has its own sparse mask, which is beneficial in inducing the desired diversity between distinct prediction heads.
> >
> > Thank you for engaging so carefully with our submission. We hope our answers have clarified the issues, and we would be grateful if you could consider an improved rating.
> >
> > [1] https://arxiv.org/abs/2305.02299

---

### Official Review · Reviewer_tnJm · 2025-10-31

**Soundness:** 2
**Presentation:** 3
**Contribution:** 2
**Rating:** 2
**Confidence:** 3

**Summary:**

This work applies dynamic sparse training to the TreeNet architecture.

**Strengths:**

- Fundamentally, this work can be viewed as a straightforward combination of DST and TreeNet, which makes it difficult to avoid potential challenges to its novelty claim. Nevertheless, assuming that no prior study has explicitly explored their integration, I would emphasize the strength of this work in providing a thorough analysis of the combination and, moreover, presenting an implementation that achieves practical speed improvements.

- The paper provides a reasonable analysis of the components of TreeNet and DST. Ultimately, what matters is whether ensemble diversity is achieved, and this is well demonstrated through the prediction disagreement analysis.

**Weaknesses:**

- It’s hard to say the presented language experiments as “LLM” experiments (title of Appendix G.1). While it’s understandable that not everyone has access to ample computational resources, the presented zero-shot reasoning results (Table 5) don’t appear particularly meaningful.

- The DST Ensemble should also benefit from DeepSparse, so it would be appropriate to include it in Figure 7. According to Table 2, the DST Ensemble achieves an accuracy of 83.3, equivalent to that of the Full Ensemble, and should therefore appear somewhere to the right of it. It would be important to check where exactly it lies relative to the NeuroTrails boundary.

**Questions:**

- Would it be possible to train the TreeNet architecture with DST in a transfer learning scenario? If so, it seems feasible to leverage a pre-trained model and conduct the language experiments accordingly.

- What about the zero-shot reasoning results corresponding to Table 16?

---

> ### Author Response · Authors · 2025-11-28
>
> Dear reviewer tnJm,
>
> Thank you for finding our paper to “provide a thorough analysis” and “achiev[ing] practical speed improvements.” We agree that the ensemble diversity is ultimately what matters, thus we appreciate that you’ve found that “this is well demonstrated through the prediction disagreement analysis.”
>
> Addressing your points and questions:
>
> W1. We have tried to refer to our language modeling experiments without using the term “LLM” throughout our work, as we indeed experiment on models of size 130M/350M (in full ensembles with M=3 this means approximately 390M / 1.05B params). Only in the subsection title of Appendix G.1 this occurred. We have adjusted it to “Parameter-Matched Language Model Study” in the pdf.
>
> W2. Thank you for pointing this out. We have measured the efficiency of the DST Ensemble to include it in Figure 7. Please see the updated graph in the current version of the pdf. As shown, NeuroTrails maintains its Pareto front across accuracy and throughput with DeepSparse.
>
> Q1. Although it is of course possible to do transfer learning with NeuroTrails, the strength of its dynamic sparse training approach comes to full fruition especially when training from scratch, as shown in [1,2].
>
> Q2. We believe we already present a thorough analysis (as you mentioned) and extensive experiments across three fields (computer vision, language modeling, reinforcement learning). We don’t think the proposed experiments would significantly add to the main insights of the paper.
>
> We appreciate your time and constructive comments. We hope our responses resolve the issues raised and would be grateful if you could consider a higher score.
>
> [1] https://arxiv.org/abs/1911.11134
> [2] https://www.nature.com/articles/s41467-018-04316-3

---

### Official Review · Reviewer_oJ41 · 2025-11-01

**Soundness:** 3
**Presentation:** 3
**Contribution:** 2
**Rating:** 4
**Confidence:** 4

**Summary:**

This paper introduces NeuroTrails, a novel and model-agnostic training paradigm for creating efficient and high-performing neural network ensembles. The core contribution lies in a unique architecture that combines a shared backbone with multiple, independent heads, where both components are trained using dynamic sparse training (DST). The authors claim this approach fosters a beneficial level of prediction diversity, which they term the "Goldilocks zone," where diversity is neither too high nor too low, leading to optimal performance. This methodology is empirically validated across a wide range of tasks in computer vision, language modeling, and reinforcement learning, demonstrating superior performance over standard ensembles and related methods while requiring significantly fewer theoretical FLOPs for inference.

**Strengths:**

- **Simplicity and Generality:** The proposed method is elegant in its simplicity. The architectural split into a shared backbone and multiple heads is intuitive and can be readily applied to various existing architectures, including CNNs (ResNet, DQN) and Transformers (LLaMA), without requiring complex new components like the routers found in Mixture-of-Experts models. The extensive experiments successfully demonstrate this model-agnostic nature.

- **Novel Conceptual Contribution:** The identification of a "Goldilocks zone" for prediction diversity is a valuable and insightful contribution to the field of ensemble learning. It challenges the common assumption that maximizing diversity is always optimal. The paper provides compelling empirical evidence for this phenomenon (Table 7) and proposes the backbone split-point as a simple yet effective hyperparameter to control this diversity, offering a practical tool for practitioners.

- **Comprehensive Empirical Validation:** The paper's claims are supported by a rigorous and extensive set of experiments across disparate domains. NeuroTrails consistently outperforms strong baselines, including full ensembles and its direct predecessor TreeNet, on standard benchmarks like CIFAR-100, ImageNet, C4, and Atari. The reported improvements in accuracy, perplexity, and out-of-distribution robustness are significant, especially when considering the reduced theoretical inference cost (FLOPs).

**Weaknesses:**

- **The Unstructured Sparsity Bottleneck (Primary Weakness):** The paper's primary weakness, and a significant threat to its practical impact, is its reliance on unstructured dynamic sparsity (RigL/SET). While this approach effectively reduces theoretical FLOPs, it is well-documented that the irregular memory access patterns of unstructured sparsity do not translate to meaningful wall-clock speedups on modern parallel hardware like GPUs without specialized libraries or hardware. The paper's own speedup analysis (Figure 7) relies on the DeepSparse engine on CPUs, which represents a limited, non-standard deployment scenario for large-scale deep learning. This stands in stark contrast to recent advancements in hardware-aware DST, such as Structured RigL (SRigL), which generates N:M semi-structured sparsity that can be directly accelerated by modern GPUs. This makes the claimed efficiency gains largely theoretical in the most common deployment environments.

- **Scalability Concerns for Foundation Models:** The NeuroTrails paradigm requires activating all M heads for every forward pass, leading to an inference cost that scales linearly with the number of heads. This "activate-all" approach poses a significant scalability challenge for future foundation models that might require hundreds or thousands of specialized components. This scaling property is markedly less efficient than that of Mixture-of-Experts (MoE) architectures, where inference cost remains constant (proportional to a small k) regardless of the total number of experts, making MoE a more viable path for massive model scaling.

- **Limited Generalizability of the "Goldilocks" Hypothesis:** While the evidence for the "Goldilocks zone" on CIFAR-100 is compelling, the paper does not sufficiently demonstrate that this is a universal principle. It remains an open question whether this phenomenon holds across different datasets (e.g., ImageNet, C4), model scales, and modalities, or if it is an artifact of the specific experimental setup presented. Further investigation is needed to establish this as a fundamental principle of ensembling.

**Questions:**

- Could you please comment on the practical inference performance (i.e., wall-clock time or throughput) of NeuroTrails on standard GPUs (e.g., NVIDIA A100) without specialized software like DeepSparse? Have you considered integrating your architectural concept with a hardware-aware DST method like Structured RigL to generate N:M sparse heads, which could potentially bridge the gap between theoretical FLOPs and actual GPU speedup?

- The "Goldilocks zone" finding is fascinating. Have you performed analyses to see if this phenomenon holds on the other domains you tested, such as ImageNet or the C4 language modeling task? How sensitive is this optimal zone to other hyperparameters, such as the sparsity ratio or the choice of DST algorithm (e.g., SET vs. RigL)?

- Given that the inference cost of NeuroTrails scales linearly with the number of heads M, how do you envision this approach scaling to future foundation models that may contain hundreds or thousands of specialized components? How does it compare to the constant-cost (top-k) scaling of MoE architectures in this large-scale regime?

---

> ### Author Response · Authors · 2025-11-28
>
> Dear reviewer oJ41,
>
> We appreciate that you have found our method to be “elegant” and “intuitive”, “consistently outperform[ing] strong baselines” on “extensive experiments.” Furthermore, we agree with your assessment that the “Goldilocks zone finding is fascinating” and a “valuable and insightful contribution” that “challenges the common assumption.”
>
> Addressing your points and questions:
>
> W1-Q1. Indeed, as discussed in Section 5.4, the FLOPs gains are currently not yet translating one-to-one to wall-clock speedups on GPUs. We show however that for edge-devices (such as the widely used AI on smartphones nowadays) the increased efficiency is in fact realized. We think this is a crucial result already.
> Moreover, using Structured RigL (SRigL) [1] is a welcome improvement for GPUs, thanks for the suggestion. We have run experiments with SRigL and show that it performs almost as well as RigL in terms of eval perplexity, while having potential to improve inference efficiency.
>
> |   Method      | Perplexity  (↓)  |
> |:-----------------|:---------------|
> |    NeuroTrails (M=3, S=0.1, RigL)    |   26.00 |
> |    NeuroTrails (M=3, S=0.1, SRigL)  |    26.27  |
>
> The inference speed on GPUs when measured simply as-is, provides the following with batch size 128.
>
> | Method          | Latency (ms)    | Throughput (samples/sec) |
> | :----------------- | :------------------- | :----------------------- |
> | Single Dense  | 20.66 ± 0.07   | 6195.73                  |
> | TreeNet          | 44.82 ± 0.12   | 2855.66                  |
> | NeuroTrails     |  44.88 ± 0.12  |  2852.25                  |
> | Full Ensemble | 62.04 ± 0.20   | 2063.09                 |
>
> We see that NeuroTrails is just as fast as TreeNet, significantly quicker than the Full Ensemble. With access to true N:M semi-structured sparsity support NeuroTrails’ speed could be improved even further, while outperforming all baselines in terms of accuracy, perplexity, and reward.
>
> W2-Q3. We agree that Mixture-of-Expert (MoE) architectures are more scalable at inference time. However, during training there is less of a difference, and the simplicity of our method ensures that we don’t require any sophisticated router mechanisms. Especially in fields such as our RL experiments these can be difficult to get right. NeuroTrails brings a simple implementation that can readily be combined with other approaches.
>
> W3-Q2. Thanks for pointing to the universality of the Goldilocks zone principle. We have run additional experiments to test the sensitivity of this optimal zone in other fields; language modeling (1 seed) and reinforcement learning (4 seeds). Per your suggestion, we also test the effect of a few hyperparameters such as the sparsity ratio of the choice of DST algorithm. The ability to tune diversity, i.e., the amount of prediction disagreement (PD), is most strongly present in the splitting point parameter $\ell$, as shown in Table 7. The other hyperparameters have some influence as well, see the results below.
>
> ## Language Modeling (LLaMA-130M on C4)
> |   Sparsity Ratio | PD (%)       | Perplexity  |
> |-----------------:|:-------------|:---------------|
> |             0      |    25.12   |        25.73 |
> |             0.05 |    26.78   |        25.67 |
> |             0.1   |    26.77   |        25.89 |
> |             0.2   |    27.41   |        26.16 |
>
>
> ## Reinforcement Learning (DQN on Asterix)
>
> ### DST method: RigL
>
> |   Sparsity Ratio | PD (%)       | Final Return      |
> |-----------------:|:-------------|:------------------|
> |             0       | 42.47 ± 4.09 | 3185.00 ± 505.00  |
> |             0.70  | 43.49 ± 1.06 | 3535.00 ± 320.00  |
> |             0.80  | 44.56 ± 3.42 | 2510.00 ± 125.00  |
> |             0.90  | 40.72 ± 0.10 | 5007.50 ± 137.50  |
> |             0.95 | 40.63 ± 1.84 | 3910.00 ± 475.00  |
> |             0.99 | 40.74 ± 3.11 | 2132.50 ± 42.50   |
>
> ### DST method: SET
> |   Sparsity Ratio | PD (%)       | Final Return      |
> |-----------------:|:-------------|:------------------|
> |             0       | 42.47 ± 4.09 | 3185.00 ± 505.00  |
> |             0.70  | 42.83 ± 0.56 | 7775.00 ± 1670.00 |
> |             0.80  | 43.86 ± 1.39 | 10087.50 ± 762.50 |
> |             0.90  | 42.91 ± 0.86 | 4437.50 ± 287.50  |
> |             0.95 | 39.96 ± 1.69 | 3790.00 ± 80.00   |
> |             0.99 | 38.25 ± 1.32 | 1525.00 ± 130.00  |
>
>
> The prediction disagreement (PD) is not impacted much by the choice of DST method. There is an interesting trend on the sparsity ratio: PD seems to be lower for very high (0.99) or very low (0) sparsity levels. For reinforcement learning, a sparsity of around 80-90% creates a well-working policy. In language modeling, such high sparsity ratios are not reachable yet, where performance is likely impacted by more than just a difference in PD.
>
> Thank you for your valuable feedback and time. We hope these clarifications address the concerns, and kindly ask you to consider increasing the rating.
>
> [1] https://arxiv.org/abs/2305.02299

---

### Note · Program_Chairs · 2026-01-17
**Submission Desk Rejected by Program Chairs**

The following references in this submission do not refer to real documents and/or have major errors in bibliographic information:

 H. Beyer. Exploratory Data Analysis. Biometrical Journal, 23(4):413-414, January 1981. ISSN 1521-4036. URL: https://doi.org/10.1002/bimj. 4710230408. (Cited on page 1)